# Gauge Equivariant Transformer

**Lingshen He**[1*]   **Yiming Dong**[1*]   **Yisen Wang**[1,2]   **Dacheng Tao**[4]   **Zhouchen Lin**[1,2,3†]

[1]Key Laboratory of Machine Perception (MOE), School of Artificial Intelligence, Peking University
[2]Institute for Artificial Intelligence, Peking University
[3]Pazhou Lab, Guangzhou 510330,
[4]JD Explore Academy, JD.com
`lingshenhe@pku.edu.cn, yimingdong_ml@outlook.com,`
`yisen.wang@pku.edu.cn, dacheng.tao@gmail.com, zlin@pku.edu.cn`

## Abstract

Attention mechanism has shown great performance and efficiency in a lot of deep learning models, in which relative position encoding plays a crucial role. However, when introducing attention to manifolds, there is no canonical local coordinate system to parameterize neighborhoods. To address this issue, we propose an equivariant transformer to make our model agnostic to the orientation of local coordinate systems (*i.e.*, gauge equivariant), which employs multi-head self-attention to jointly incorporate both position-based and content-based information. To enhance expressive ability, we adopt regular field of cyclic groups as feature fields in intermediate layers, and propose a novel method to parallel transport the feature vectors in these fields. In addition, we project the position vector of each point onto its local coordinate system to disentangle the orientation of the coordinate system in ambient space (*i.e.*, global coordinate system), achieving rotation invariance. To the best of our knowledge, we are the first to introduce gauge equivariance to self-attention, thus name our model Gauge Equivariant Transformer (GET), which can be efficiently implemented on triangle meshes. Extensive experiments show that GET achieves state-of-the-art performance on two common recognition tasks.

## 1   Introduction

Recently, Transformer has dominated the area of Natural Language Processing [48]. Its key advantage over previous methods is its ability to attend to the most relevant part in a given context. This is largely attributed to its self-attention operator, which computes the similarity between representations of words in sequences in the form of attention scores. Because of the superiority, researchers start to apply Transformer to other learning areas, including Computer Vision [26, 53, 16, 59] and Graphs [49].

In this work, we aim at applying Transformer to manifolds. Unlike regular data, such as images, where each neighbor owns a clearly quantified relative position to its center in a canonical coordinate system, irregular data do not have a uniquely defined local coordinate system for the neighbors, resulting in the problem of orientation ambiguity, which directly obstructs the Transformer to numerically intake the relative position information.

Several works have been proposed to deal with the rotation ambiguity problem, in which a promising way is to exploit gauge equivariance. While most of them are not rotation invariant to global

---

*Work was done during an internship at JD Explore Academy.

*Equal contribution. Sorted by tossing the coin.

†Corresponding author.

coordinate system, all of them are established on convolution, *i.e.*, equal attention to neighboring points and neglect to content-based information. So it is desirable to propose a gauge equivariant transformer with the support of rotation invariance.

In this paper, we propose Gauge Equivariant Transformer, named GET for short, which employs multi-head self-attention to simultaneously utilize position-based and content-based information, and is both gauge equivariant and rotation invariant. To achieve rotation invariance, we first project $xyz$ coordinates in a global coordinate system onto a local coordinate frame, and then design equivariant transformers to overcome the orientation ambiguity problem of local coordinate systems. We adopt the regular field proposed in [13] as feature fields of intermediate layers, since the representation of regular field commutes with element-wise activation functions. After that, we propose a novel method to accommodate parallel transport of feature vectors in regular field with any rotation angles. Since we adopt regular fields in intermediate layers, we make a relaxation such that they are equivariant only for gauge transformations of angles that are multiples of $2\pi/N$. Exact equivariance can be guaranteed for gauge transformations at multiples of $2\pi/N$, and an equivariance error bound can be obtained for all other angles. In experiments, our model shows better performance and greater parameter efficiency than all baseline methods. Our contributions can be summarized as follows:

- We propose GET, which incorporates attention and achieves both gauge equivariance and rotation invariance with superior expressive power. GET is mathematically proven to be exactly equivariant on angles that are multiples of $2\pi/N(N \in \mathbb{N}^*)$, and an equivariance error bound is derived for other angles to guarantee the overall approximate equivariance property.

- We carefully design the model input to ensure that it is irrelevant to the global coordinate system, only depending on the choice of gauge. Our model achieves rotation invariance with the assistance of gauge equivariance.

- We propose a novel method to parallel transport the feature vectors in the regular field by extending the representation of a cyclic group to any angle rotation group. Compared to previous methods using truncation or interpolation, our extension can preserve more geometric information.

- We elevate the model performance by designing a new approach which incorporates Taylor expansion in solving the equivariance constraint, which has a better approximation ability in local neighborhoods.

- We confirm the superiority of our model via extensive experiments. Our model outperforms the HSN model on the SHREC dataset by $3.1\%$ accuracy, and outperforms the MeshCNN model on the Human Body Segmentation dataset by $0.3\%$ accuracy with much fewer parameters, presenting state-of-the-art performance.

## 2 Related Work

**Geometric Deep Learning.** Geometric deep learning is an emerging field concerning with adapting neural networks to various data types [7], especially on irregular data. For research works on modeling curved surfaces, common methods include view-based methods [46, 61, 51] and volumetric methods [32, 40, 52]. To boost efficiency, some works define convolution on point clouds directly [37, 38], but they are vulnerable to pose change since the coordinate inputs are dependent of the global coordinate system. So it is highly desired to develop models that solely intake geometric information of surfaces.

Approaches that merely utilize intrinsic information of surfaces are called intrinsic methods. They use local parameterization to assign each neighboring point with a coordinate for information aggregation. A seminal work is Geodesic CNNs [31], which uses an exponential map to parameterize each local neighborhood and takes the maximum response across multiple choices of local coordinate orientation. While taking the maximum response direction discards the orientation information of feature maps, as an alternative, aligning local coordinate with principle curvature direction is another approach to deal with the ambiguity problem [33, 6]. But this approach can only be applied in limited cases as the curvature direction may be ill-defined at some points or even areas of curved surfaces. MDGCNN [35] and PFCNN [60] describes features by the so called directional functions. But both of them adopt scalar equivariant kernels, resulting in limited expressive power.

**Equivariant Deep Learning.** Success of CNNs has been attributed to translation equivariance, which inspires researchers to implement more powerful equivariant models, including equivariance of planar rotation [12, 15, 13, 58, 54, 44, 25], 3D space rotation [57, 19, 47, 36, 55, 35, 28, 2, 39], sphere rotation [9, 17, 34, 18], and so on. All above works are about equivariance on homogeneous space [29, 10]. Cohen et al. [11] further extend equivariance to manifolds, in which they identify a new type of equivariance called gauge equivariance. The models in [56, 14] are successful extensions of gauge equivariant CNNs on mesh surfaces. However, their model suffers from changes in the orientation of global coordinate system.

Also, there are works proposed for equivariant attention. Romero et al. [43] propose co-attentive equivariant networks, which effectively attends to co-occurring transformations. Romero et al. [41] further propose attentive group equivariant convolutional networks. Besides this, transformers have also been applied to group equivariant networks, where Fuchs et al. [21] do so via irreducible representations, Hutchinson et al. [27] via Lie algebra, and Romero et al. [42] via generalization of position encodings. All the models above are equivariant to symmetric groups, while currently gauge equivariant attention is still lacking.

## 3 Preliminaries

Unlike regular data, in which coordinates (or pixels) are aligned in a global frame, there is no such specific frame on general manifolds. To begin with, we briefly review and define some mathematical concepts.

### 3.1 Basic Definitions

We restrict our attention to 2D manifolds in 3D Euclidean space. Consider a 2D smooth orientable manifold $M$. For a point $p$ in $M$, denote its *tangent plane* as $T_pM$. Each point in $T_pM$ can be associated with a coordinate by specifying a coordinate system. Namely, we can parameterize the tangent plane $T_pM$ with a pointwise linear mapping $w_p : \mathbb{R}^2 \to T_pM$, which is defined as the *gauge* $w$ at point $p$ [11]. The gauge of manifold $M$ is the set containing gauges at every point in $M$.

For planar data, a feature map is the set of features located at different positions on a plane. Similarly, a *feature field* on a surface is a set of geometric quantities at different positions of the surface. Note that these two concepts are similar but not the same. From the perspective of geometric deep learning, a *feature map* is defined as numerical values of geometric quantities that may be gauge dependent, while a feature field refers to geometric quantities themselves that are gauge independent. For example, each point of the surface can be assigned with a tangent vector as its feature vector, all of which form a feature field. As is shown in Figure 1, the tangent vector $v$ itself is a *geometric quantity*, which stays the same regardless of arbitrary gauge selection but takes different numerical values in different gauges following an underlying rule. We use $f$ to denote the feature field of a manifold, $f_w : M \to \mathbb{R}^n$ denotes the feature map under the gauge $w$ and $f_w(p)$ denotes the feature map evaluated at point $p$.

Different gauges can be linked by gauge transformations. The *gauge transformation* at point $p$ is a frame transformation: $g_p \in SO(2)$, where $SO(2)$ is the *special orthogonal group* consisting of all 2D rotation transformation matrices. A new gauge $w'_p$ can be produced by applying gauge transformation $g_p$ to the original gauge $w_p$, *i.e.*, $w'_p = g_p \cdot w_p$. Gauge transformation is usually characterized by group representations. *Group representation* is a mapping $\rho : G \to GL(n, \mathbb{R})$ where $GL(n, \mathbb{R})$ is the group of invertible $n \times n$ matrices, and $\rho$ meets the condition $\rho(g_1)\rho(g_2) = \rho(g_1 g_2)$, where $g_1, g_2 \in G$ are the elements of the group, $g_1 g_2$ are element product defined on the group, and $\rho(g_1)\rho(g_2)$ is matrix multiplication. Therefore, after applying the gauge transformation $g_p$, the feature vector value $f_w(p)$ transforms to $f_{w'}(p) = \rho\left(g_p^{-1}\right) f_w(p)$. Here $\rho$ is a group representation of $SO(2)$ which is called the *type* of the feature vector. If all the feature vectors share the same type $\rho$, the feature field is called a *$\rho$-field* and $\rho$ is called the representation type of the field. The above definitions can also be at the manifold level, *i.e.*, $f_{w'} = \rho(g^{-1})f_w$. The notation $k\rho$, where $k$ is a positive integer, refers to the group representation whose output is $k$-blocks diagonal matrix with each block equals to $\rho$. In particular, if the representation of a feature field is $\rho(g) = 1$, then the feature field becomes *scalar field*, denoted as $\rho_0$.

### 3.2 Gauge Equivariance

For a function $\phi$, its input is a feature map $f_w$, where $f$ is a $\rho_{in}$-field, in order to make $\phi$ gauge equivariant, and its output $\tilde{f}_w$ should be a feature map, where $\tilde{f}$ is a $\rho_{out}$-field. When $\phi$ is a layer of a neural network, gauge equivariance implies that $\phi$ does not rely on the gauge in the forward process.

Suppose that there are two gauges $w$ and $w'$ linked by a gauge transformation $g$: $w' = g \cdot w$, we have $f_{w'} = \rho_{in}(g^{-1})f_w$ since $f$ is a $\rho_{in}$-field. *Gauge equivariance* means that the outputs $\tilde{f}_w = \phi[f_w]$ and $\tilde{f}_{w'} = \phi[f_{w'}]$ are linked by the $\rho_{out}$ representation of the same transformation $g$, *i.e.* $\tilde{f}_{w'} = \rho_{out}(g^{-1})\tilde{f}_w$. Finally, we get:

$$\rho_{out}(g^{-1})\phi[f_w] = \phi\left[\rho_{in}(g^{-1})f_w\right]. \tag{1}$$

To sum up, a function $\phi$ is gauge equivariant if the above equation always holds for any feature field $f$, gauge $w$ and transformation $g$.

### 3.3 Riemannian Exponential Map

Transformers require encoding the relative position to propagate information. Note that in images, there is still a local point parameterization, which is so natural that one even does not realize it. For general manifolds, it is non-trivial to establish a parameterization criterion, at least in the local frame. Among many charting-based methods, the mostly used one is the *Riemannian exponential map* $\exp_p : T_pM \to M$ at point $p$, which is a mapping from the tangent plane to the surface. For a coordinate vector $v \in T_pM$, the output of the Riemannian exponential map is obtained by moving the point $p$ in the direction $v$ along the geodesic curve with a distance of $\|v\|$. Denoting the arrival point as $q$, we have $\exp_p(v) = q$. Figure 1 visualizes the exponential map as well as some basic definitions introduced in Section 3.1. According to the inverse function theorem, $\exp_p$ is a local diffeomorphism so can avoid metric distortion at the point $p$. The inverse of Riemannian exponential map is the *logarithmic map* $\log_p : M \to T_pM$. Under the gauge $w_p$, every point $q$ in the neighborhood of $p$ is associated with coordinate $w_p^{-1} \cdot \log_p(q)$.

### 3.4 Parallel Transport

The self-attention operation is essentially an aggregation of local neighboring features. However, the feature vectors of different points are from different spaces, thus they need to be parallel transported to the same feature space before being processed. For a tangent vector $s$ at point $q$, we parallel transport it along the geodesic curve to another point $p$ with respect to Levi-Civita connection [8], which preserves the norm of the vector. Levi-Civita connection is an isometry from $T_qM$ to $T_pM$ and determines the parallel transport of $s$, see Figure 2. In a gauge $w$, the parallel transport of tangent vector corresponds to a 2D rotation $g_{q \to p}^w \in SO(2)$ which contains the relative orientation of gauges in the neighborhood. For a general feature vector of $\rho$ type, parallel transport can be expressed as $s'_w = \rho(g_{q \to p}^w)s_w$.

### 3.5 Self-attention

Attention enables the model to selectively concentrate on the most relevant parts based on their content information [48, 53, 4, 22]. Consider a set of tokens $t = \{t_1, t_2, \ldots, t_T\}$, where $t_i \in \mathbb{R}^F$. Attention is composed of three parts, namely *query*, *key* and *value*, denoted by $Q : \mathbb{R}^F \to \mathbb{R}^{F_Q}$, $K : \mathbb{R}^F \to \mathbb{R}^{F_K}$, and $V : \mathbb{R}^F \to \mathbb{R}^{F_V}$, respectively. When $Q$, $K$ and $V$ are from the same source, it is called *self-attention*. When there are multiple sets of $Q$, $K$ and $V$'s, it becomes *multi-head attention*.

The output of a multi-head self-attention transformer at node $i$ is the linear transformation of the concatenation of the outputs at all the heads:

$$\text{MHSA}(t)_i = W_M\left(\Big\|_h \text{SA}(t)_i^{(h)}\right), \tag{2}$$

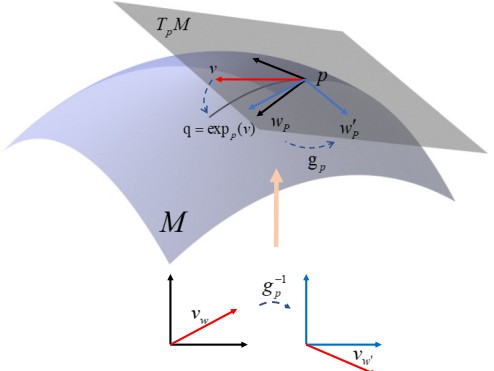
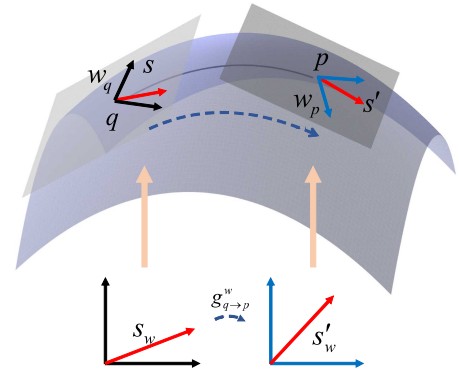

Figure 1: Illustration of basic definitions and Riemannian exponential map. Here, $w_p$ (black) and $w'_p$ (blue) are two gauges on the tangent plane $T_pM$ and they are linked by the gauge transformation $g_p$. The coordinate of $v$ takes different numerical values under $w_p$ and $w'_p$, as is illustrated in lower part. The exponential map assigns each vector $v$ in $T_pM$ with corresponding point $q$ on the surface $M$.

Figure 2: Parallel transport. The tangent vector $s$ is parallel transported from $q$ to $p$, resulting in a new vector $s'$ at point $p$. The numerical value change imposed by parallel transport is jointly determined by the geometric property of the surface, the Levi-Civita connection and the underlying gauge $w$.

where $\parallel$ is the vector concatenation operator. The single head attention output at head $h$ is

$$\mathrm{SA}(t)_i^{(h)} = \sum_{j=1}^{T} \alpha_{ij}^{(h)} V^{(h)}(t_j), \tag{3}$$

where $V^{(h)}$ is the value function at the head $h$, and $\alpha_{ij}^{(h)}$ is attention score computed by

$$\alpha_{ij}^{(h)} = \frac{S(K^{(h)}(t_i), Q^{(h)}(t_j))}{\sum_{j'=1}^{T} S(K^{(h)}(t_i), Q^{(h)}(t_{j'}))}, \tag{4}$$

where $K^{(h)}$, $Q^{(h)}$ and $S$ are the key function, query function and score function, respectively.

## 4 The Proposed GET

### 4.1 Gauge Equivariant Self-Attention Layers

Suppose that the dimensions of input feature field $f$ and output feature field $\tilde{f}$ are $C_{in}$ and $C_{out}$, respectively. We define the gauge equivariant multi-head self-attention output at point $p$ under the gauge $w$ as

$$\tilde{f}_w(p) = \mathrm{MHSA}(f)_w(p) = W_M \left( \parallel_h \mathrm{SA}(f)_w^{(h)}(p) \right), \tag{5}$$

where $W_M$ is the linear transformation matrix. At the head $h$, the output is defined as

$$\mathrm{SA}(f)_w^{(h)}(p) = \int_{\|u\|<\sigma} \alpha(f)_{p,q_u}^{(h)} V_u^{(h)}(f'_w(q_u)) \mathrm{d}u, \tag{6}$$

where $u = (u_1, u_2)^T \in \mathbb{R}^2$, $q_u = \exp_p w_p(u)$, $f'_w(q_u)$ is the numerical value of parallel transported feature vector from point $q_u$ to point $p$ under the gauge $w$, and $V_u$ is the value function incorporating the position information $u$ through an encoder matrix $W_V(u) \in \mathbb{R}^{C_{out} \times C_{in}}$, *i.e.*

$$f'_w(q_u) = \rho_{in}(g_{q_u \to p}^w) f_w(q_u), \quad V_u(f'_w(q_u)) = W_V(u) f'_w(q_u). \tag{7}$$

$\alpha$ is the attention score incorporing the content information, and is computed as:

$$\alpha(f)_{p,q_u}^{(h)} = \frac{S(K^{(h)}(f_w(p)), Q^{(h)}(f'_w(q_u)))}{\int_{\|v\|<\sigma} S(K^{(h)}(f_w(p)), Q^{(h)}(f'_w(q_v)))\mathrm{d}v}. \tag{8}$$

We propose to enforce the attention score to be gauge invariant and the value function to be gauge equivariant, to make the attention layer gauge equivariant. The details of constructing them are presented in Sections 4.3 and 4.4, respectively.

## 4.2   Extension of Regular Representation

In our model, the feature fields in the intermediate layers are all regular fields (*i.e.*, whose type is regular representation). Regular representation is a special type of group representation of $C_N$. If we use $\Theta_k$ to denote the rotation matrix with angle of $k \cdot 2\pi/N$, then $C_N$ can be expressed as $C_N = \{\Theta_0, \Theta_1, \cdots, \Theta_{N-1}\}$. For $k = 0, 1, \cdots, N-1$, the regular representation $\rho_{reg}^{C_N}(\Theta_k)$ is an $N \times N$ cyclic permutation matrix which shifts the coordinates of feature vectors by $k$ steps.

Regular representation provides transformation matrices when rotating by angles of multiples of $2\pi/N$, but feature vectors can go through any rotation in $SO(2)$ during parallel transport. Figure 3 illustrates this issue by giving an example in $\mathbb{R}^5$ with respect to $\rho_{reg}^{C_5}$. We propose to extend the regular representation of $C_N$ by finding an orthogonal representation $\tilde{\rho}_N$ of $SO(2)$, such that it behaves the same as regular representation for any element in $C_N$, *i.e.*

$$\forall \Theta \in C_N, \tilde{\rho}_N(\Theta) = \rho_{reg}^{C_N}(\Theta). \tag{9}$$

As $\rho_{reg}^{C_N}$ takes different forms between odd values and even values of $N$, Theorem 1 shows that only odd $N$'s are vaild in our model.

**Theorem 1** *(i) If $N$ is even, there is no such real representation $\tilde{\rho}_N$ of $SO(2)$ that satisfies Eqn. (9). (ii) If $N$ is odd, there is a unique representation $\tilde{\rho}_N$ of $SO(2)$ that satisfies Eqn. (9). (iii) The representation $\tilde{\rho}_N$ in (ii) is an orthogonal representation.*

Here we only show our method for constructing $\tilde{\rho}_N$ in Theorem 1. According to group representation theory, regular representation $\rho_{reg}^{C_N}$ can be decomposed into irreducible representations (irrep for short), *i.e.*,

$$\rho_{reg}^{C_N}(\Theta) = A \operatorname{diag}\left(\varphi_0(\Theta), \varphi_1(\Theta), \cdots, \varphi_{\frac{N-1}{2}}(\Theta)\right) A^{-1}, \tag{10}$$

where $\varphi_0, \cdots, \varphi_{(N-1)/2}$ are the irreps of $C_N$, and $A \in GL(N, \mathbb{R})$. The irreps of $C_N$ take the following form for odd $N$:

$$\forall \Theta \in C_N, \varphi_0(\Theta) = 1, \varphi_k(\Theta) = \begin{bmatrix} \cos(k\theta) & -\sin(k\theta) \\ \sin(k\theta) & \cos(k\theta) \end{bmatrix}, \tag{11}$$

where $\theta \in [0, 2\pi)$ is the rotation angle of the matrix $\Theta$, *i.e.*

$$\Theta = \begin{bmatrix} \cos\theta & -\sin\theta \\ \sin\theta & \cos\theta \end{bmatrix}, \tag{12}$$

and $k = 1, \cdots, \frac{N-1}{2}$. We extend the irreps to $SO(2)$ as

$$\forall \Theta \in SO(2), \tilde{\varphi}_0(\Theta) = 1, \tilde{\varphi}_k(\Theta) = \begin{bmatrix} \cos(k\theta) & -\sin(k\theta) \\ \sin(k\theta) & \cos(k\theta) \end{bmatrix}, \tag{13}$$

where $k = 1, \cdots, \frac{N-1}{2}$. By substituting the $\varphi$'s in Eqn. (10) with $\tilde{\varphi}$'s, we get that for $\forall \theta \in SO(2)$,

$$\tilde{\rho}_N(\Theta) = A \operatorname{diag}\left(\tilde{\varphi}_0(\Theta), \tilde{\varphi}_1(\Theta), \cdots, \tilde{\varphi}_{\frac{N-1}{2}}(\Theta)\right) A^{-1}. \tag{14}$$

Obviously the representation $\tilde{\rho}_N$ satisfies the condition Eqn. (9). In this way, one can apply $\tilde{\rho}_N(g_{q \to p}^w)$ to feature vector of regular field during parallel transport.

### 4.3 Gauge Equivariant Value Function

Inspired by [11], we choose the value function to be the numerical value of the parallel transported feature vector multiplied by the value encoding matrix. For the value function to be gauge equivariant, the necessary and sufficient condition is that Eqn. (15) always holds for any $\Theta \in SO(2)$:

$$W_V(\Theta^{-1}u) = \rho_{out}(\Theta^{-1})W_V(u)\rho_{in}(\Theta). \tag{15}$$

We propose a practical method to solve Eqn. (15). We first expand the $W_v$ into taylor series:

$$W_V(u) = W_0 + W_1 u_1 + W_2 u_2 + W_3 u_1^2 + W_4 u_1 u_2 + W_5 u_2^2 + \cdots , \tag{16}$$

where $W_i \in \mathbb{R}^{C_{out} \times C_{in}} (i = 0, 1, \cdots)$ is the Taylor coefficient. Since we adopt regular representation in this paper, Eqn. (15) only needs to hold for $\Theta \in C_N$. Plugging Eqn. (16) into Eqn. (15), by comparing the coefficients, $W_i$'s need to satisfy that for any $\Theta \in C_N$,

$$W_0 = \rho_{out}(\Theta^{-1})W_0\rho_{in}(\Theta), \tag{17a}$$

$$\cos(\theta)W_1 - \sin(\theta)W_2 = \rho_{out}(\Theta^{-1})W_1\rho_{in}(\Theta), \tag{17b}$$

$$\sin(\theta)W_1 + \cos(\theta)W_2 = \rho_{out}(\Theta^{-1})W_2\rho_{in}(\Theta), \tag{17c}$$

$$\cdots .$$

To deal with the issue of having infinite terms in Eqn. (16), we may bypass it by simply truncating the Taylor series. We use the second order Taylor expansion and omit higher order terms, *i.e.*,

$$W_V(u) \triangleq W_0 + W_1 u_1 + W_2 u_2 + W_3 u_1^2 + W_4 u_1 u_2 + W_5 u_2^2. \tag{18}$$

It is worth emphasizing that making truncations does not affect the equivariance property in the slightest, as the equations in (17) show the coupling characteristics.

Eqn. (17a) is the constraint on $W_0$ in the order 0, Eqn. (17b) and Eqn. (17c) are the constraints on $W_1$ and $W_2$ in order 1, and there are three more equations in Eqn. (17) constraining on $W_3$, $W_4$ and $W_5$ in the order 2. We can see that only the $W_i$'s in the same order are coupled together. This coupling property allows us not only to solve the equations in (17) in separate groups, but also to make truncations in Eqn. (16) without affecting the equivariance property.

After truncation, we can get a set of solution bases of Taylor coefficients $\{\tilde{W}^{(1)}, \cdots, \tilde{W}^{(m)}\}$ by solving the first six linear equations in (17) which are separated into three independent groups, where $m$ is the dimension of solution space. Each $\tilde{W}^{(i)}$ is a tuple consisting of six components, $\tilde{W}_0^{(i)}, \cdots, \tilde{W}_5^{(i)}$. The details in solving linear equations are provided in supplementary materials. Then, the equivariant matrix basis $W^{(i)}$ has the following form:

$$W^{(i)}(u) = \tilde{W}_0^{(i)} + \tilde{W}_1^{(i)}u_1 + \tilde{W}_2^{(i)}u_2 + \tilde{W}_3^{(i)}u_1^2 + \tilde{W}_4^{(i)}u_1 u_2 + \tilde{W}_5^{(i)}u_2^2, \tag{19}$$

which satisfies Eqn. (15) for all $u$. Their linear combination, $\sum c_i W^{(i)}$, still meets Eqn. (15) and $c_i$'s can be set as learnable parameters during training. With $W_V = \sum c_i W^{(i)}$, the value function in Eqn. (7) is exactly equivariant to gauge transformations at multiples of $2\pi/N$.

*It is remarkable that our method of solving the equivariance constraint Eqn. (15) is very general, as the solution process can be applied to any groups, including $\rho_{in}$ and $\rho_{out}$. Especially, it can avoid solving analytic solutions when the group is very complex, like when the case is that the group is a high dimensional orthogonal group.* In addition, compared to Fourier series used in [54], Taylor series is a better approximation in local neighborhoods. The omitted Taylor terms in Eqn. (18) is $O(\sigma^3)$, which is negligible when the radius $\sigma$ is small enough. So GET could achieve the same performance with fewer parameters. In addition, we can avoid selecting radial profiles that introduce extra hyperparameters.

### 4.4 Gauge Invariant Attention Score

In implementation, the manifold is discretized to mesh for computer processing. The discretization details are provided in supplementary materials. Here we set the key and query function to be structurally the same as Graph Attention Network [49], *i.e.*, $K^{(h)}(f_w(p)) = W_K^{(h)} f_w(p)$,

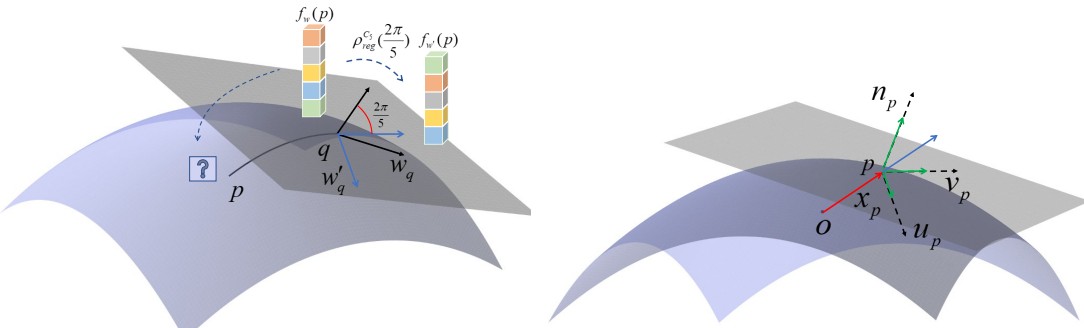

Figure 3: Illustration for the reason of extension. $f(q)$ is a feature vector of type $\rho_{reg}^{C_5}$, which takes numerical value $f_w(q) \in \mathbb{R}^5$ under gauge $w_q$. Applying a gauge transformation with angle $2\pi/5$ to $w'_q$, $f(q)$ takes another value $f_{w'}(q)$, which is a permutation of $f_w(q)$. The problem here is what value does $f(q)$ takes after it is parallel transported to point $p$.

Figure 4: Local coordinate projection. $x_p$ is the position vector in the global coordinate system marked in red. For better illustration it is moved to the local coordinate system, marked in blue. In the local coordinate system $x_p$ is projected onto the directions of $u_p$, $v_p$ and $n_p$, respectively, and the lengths of three directed line segments (in green) form the input $X_p$.

$Q^{(h)}(f'_w(q_u)) = W_Q^{(h)} f'_w(q_u)$, where $W_K^{(h)} \in \mathbb{R}^{N \times C_{in}}$, $W_Q^{(h)} \in \mathbb{R}^{N \times C_{in}}$. The score function is structurally similar to [49], which takes the following form:

$$S(K(\cdot), Q(\cdot)) = P(ReLU(K(\cdot) + Q(\cdot))). \tag{20}$$

Here, $ReLU$ is the Nonlinear Rectified Unit acting on each element of the $N$ dimensions, and $P : \mathbb{R}^N \to \mathbb{R}$ is the average pooling function. The linear transformation matrices $W_K$ and $W_Q$ are required to satisfy the constraint in Eqn. (17a) on $C_N$ for $K$ and $Q$ to be gauge equivariant. After activation and pooling, the final attention score is gauge invariant.

With the gauge invariant attention score and gauge equivariant value function, the single head attention Eqn. (6) is gauge equivariant. For the multi-head attention to be gauge equivariant, the transformation matrix $W_M$ also needs to satisfy Eqn. (17a).

### 4.5 Rotation Invariance

The rotation invariance property of GET is accomplished by constructing a local coordinate system for every point and making use of the gauge equivariance property. As is shown in Figure 4, assuming that $x_p$ is the coordinate vector of $p \in M$ in the global coordinate system, $n_p$ is the corresponding normal vector, and the gauge $w_p$ is ascertained by principal axes $u_p$ and $v_p$. By projecting the raw data $x_p$ onto the local coordinate system, we get the local coordinate of point $p$: $X_p = (\langle x_p, u_p \rangle, \langle x_p, v_p \rangle, \langle x_p, n_p \rangle)$, which relies on $w_p$ but is invariant to the choice of global coordinate system. The insight is that $X$ is actually a feature map whose corresponding feature field is associated with representation $\rho_{local}$ as:

$$\rho_{local}(\Theta) = \begin{bmatrix} \cos\theta & -\sin\theta & 0 \\ \sin\theta & \cos\theta & 0 \\ 0 & 0 & 1 \end{bmatrix}. \tag{21}$$

If we feed the local coordinates into an $SO(2)$ gauge equivariant model whose outputs are scalar fields, the result will be $SO(3)$ rotation invariant.

### 4.6 Error Analysis

Following the conventions, GET stacks multiple self-attention layers with ReLU activation functions. Even if discretized on triangle meshes, GET is still exactly equivariant to gauge transformations at angles that are multiples of $2\pi/N$.

**Theorem 2** *Assume a GET $\psi$, whose types of input, intermediate, and output feature fields are $\rho_{local}$, $k_i \rho_{reg}^{C_N}$ and $\rho_0$, respectively, where $k_i$ is the number of regular fields in the $i^{th}$ intermediate feature*

*field. Denote $f$ as the input feature field on triangle mesh $M$, and the norm of the feature map is bounded by constant $C$. Gauges $w$ and $w'$ are linked by transformation $g$. Further suppose that $\psi$ is Lipschitz continuous with constant L, then we have:*

*(i) If $g_p \in C_N$ for every mesh vertex $p \in M$, then $\psi(f_w) = \psi(f_{w'})$.*

*(ii) For general $g_p \in SO(2)$, we have $\|\psi(f_w) - \psi(f_{w'})\| \leq \frac{\pi L}{N} C$.*

Theorem 2 provides a bound for gauge transformation with respect to any angles. Compared to non-equivariant models, GET decreases the equivariance error by a factor of $1/N$. In experiments, we empirically show that the performance of our model increases as $N$ increases.

## 5 Experiments

We conduct extensive experiments to evaluate the effectiveness of our model. We test the performance of our model on two deformable domain tasks, and conduct parameter sensitivity analysis and several ablation studies to make a comprehensive evaluation. Note that we use data preprocessing to precompute some useful preliminary values in order to save training time. The details of preprocessing can be found in supplementary materials.

### 5.1 Shape Classification

Our model used here is lightweight but powerful. The details of the architecture and training settings are provided in supplementary materials. Under the same setting, we compare our model with HSN [56], MeshCNN [24], GWCNN [20], GI [45] and MDGCNN [35], whose results are cited in [56]. As is shown in Table 5.2, our model achieves state-of-the-art performance on this dataset. GET significantly improves the previous state-of-the-art model HSN by $3.1\%$ in classification accuracy. This may attribute to the attention mechanism and the intrinsic rotation invariance of our model, while all other models are CNNs and directly accepts the raw coordinates $xyz$ as input. Also, HSN is the most parameter efficient model among the models we compared with. Our model consumes only $1/7$ parameters of HSN (11K vs. 78K).

### 5.2 Shape Segmentation

A widely used task in 3D shape segmentation is Human Body Segmentation [30], in which the model is to predict body-part annotation for each sampled point. The dataset consists of 370 training models from MIT [50], FAUST [5], Adobe Fuse [1] and SCAPE [3] and 18 test models from SHREC07 [23]. The readers may refer to supplementary materials for details of neural network architecture and hyperparameters.

Table 2 reports the percentage of correctly classified vertices across all samples in the test set. The results of comparing models are cited from [56], [60] and [35]. Our model outperforms all these models in the segmentation task. GET consumes only about $1/15$ parameters compared with MeshCNN (148K vs. 2.28M) but achieves higher performance.

Table 1: Model results on the SHREC dataset. GET performs the best without rotation data augmentation. The models trained without rotation augmentation are rotation invariant intrinsically.

| Model | Rotation Aug. | Acc. (%) |
|---|---|---|
| MDGCNN[35] | ✓ | 82.2 |
| GI[45] | ✓ | 88.6 |
| GWCNN[20] | ✓ | 90.3 |
| MeshCNN[24] | ✗ | 91.0 |
| HSN[56] | ✓ | 96.1 |
| **GET (Ours)** | ✗ | **99.2** |

Table 2: Segmentation results on the Human Body Segmentation dataset. Our GET performs the best even without data augmentation by rotations.

| Model | Rotation Aug. | Acc. (%) |
|---|---|---|
| MDGCNN [35] | ✓ | 89.5 |
| PointNet++ [38] | ✓ | 90.8 |
| HSN [56] | ✓ | 91.1 |
| PFCNN [60] | ✗ | 91.5 |
| MeshCNN [24] | ✗ | 92.3 |
| **GET (Ours)** | ✗ | **92.6** |

## 5.3 Parameter Sensitivity

**Order of the Group $C_N$.** The hyperparameter $N$ is a key factor to the model equivariance since it controls both the dimension of regular field and the number of angles at which the our model is exactly equivariant. Also, Theorem 2 asserts that the equivariance error is bounded by a factor of $1/N$ compared to non-equivariant models. Here we study the effect of $N$ on model accuracy while keeping parameter numbers roughly the same. The results of the Human Body Segmentation dataset with different $N$'s are shown in Table 3. We can see that the model performance improves considerably as $N$ increases and stabilizes finally.

Table 3: Model accuracy and the number of parameters in the Human Body Segmentation task with respect to different $N$'s.

| $N$ | 3 | 5 | 7 | 9 (Chosen) | 11 |
|---|---|---|---|---|---|
| Acc. (%) | 91.2 | 92.0 | 92.4 | **92.6** | 92.5 |
| # Params. | 153K | 149K | 149K | 148K | 156K |

## 5.4 Ablation Study

In this section, we perform a series of ablation studies to analyze individual parts of our model. All the experiments are carried out on the Human Body Segmentation dataset under the same setting as in Section 5.2. We evaluate the effectiveness of gauge equivariance, attention, local coordinate and parallel transport method, with the latter two experiments provided in supplementary materials.

**Gauge Equivariance and Attention.** To confirm the effectiveness of gauge equivariance property and attention mechanism, we design two baseline models with one not equivariant and the other based on convolution. For the non-equivariant baseline, we use Graph Attention Networks [49]. For the convolution-based model, we adopt a similar architecture as GET.

Table 4: Model accuracy in the Human Body Segmentation task with two baselines without gauge equivariance and attention, respectively.

| Model | Gauge Equivariance | Attention | Acc. (%) |
|---|---|---|---|
| **GET** | ✓ | ✓ | **92.6** |
| Baseline 1 | | ✓ | 81.1 |
| Baseline 2 | ✓ | | 92.3 |

Table 4 shows that GET both benefits from the power of gauge equivariance and attention. We can see that both properties do contribute to the superiority of the model performance.

## 6 Conclusion

We propose GET, which firstly incorporates attention in gauge equivariance. GET introduces a new input, which is invariant to rotation of the global coordinate system. GET employs a new parallel transport approach, which is plausible for parallel transport between any two points. GET utilizes Taylor expansion in solving equivariant constraints, achieving better approximation ability. GET achieves state-of-the-art performances on several tasks and is efficient among the baselines.

## Acknowledgment

Zhouchen Lin was supported by the NSF China (No.s 61625301 and 61731018), NSFC Tianyuan Fund for Mathematics (No. 12026606) and Project 2020BD006 supported by PKU-Baidu Fund. Yisen Wang is partially supported by the National Natural Science Foundation of China under Grant 62006153, and Project 2020BD006 supported by PKU-Baidu Fund.

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
