# Gauge Equivariant Transformer
# Supplementary Materials

## 1 A  Our Discretization Method

2 In practice, the manifold is often represented by triangle mesh: a collection of vertices, edges and
3 faces. Since most concepts defined on manifolds in this paper can be naturally extended to meshes,
4 we do not repeat all of them here but only focus on the part with significant differences.

5 On meshes, the processing of single head self-attention is discretized into following form:

$$\text{SA}(f)_w^{(h)}(p) = \sum_{q \in \mathcal{N}_p} \alpha(f)_{p,q}^{(h)} V_{u_q}^{(h)}(f_w'(q)), \tag{21}$$

6 where $u_q = w_p^{-1} \log_p(q)$, $f_w'(q) = \rho_{in}(g_{q\to p}^w) f_w(q)$, $V_{u_q}(f_w'(q)) = W_V(u_q) f_w'(q)$, and

$$\alpha(f)_{p,q}^{(h)} = \frac{S(K^{(h)}(f_w(p)), Q^{(h)}(f_w'(q)))}{\sum_{q' \in \mathcal{N}(p)} S(K^{(h)}(f_w(p)), Q^{(h)}(f_w'(q')))}. \tag{22}$$

7 In implementation, the rotation induced by parallel transport $g_{q\to p}^w$ and the logarithmic map are
8 computed by the Vector Heat Method [7].

## 9 B  Proofs of the Theorems

### 10 B.1  Proof of Theorem 1

11 **Theorem 1** *(i) If $N$ is even, there is no such real representation $\tilde{\rho}_N$ of $SO(2)$ that satisfies Eqn.*
12 *(9). (ii) If $N$ is odd, there is a unique representation $\tilde{\rho}_N$ of $SO(2)$ that satisfies Eqn. (9). (iii) The*
13 *representation $\tilde{\rho}_N$ in (ii) is an orthogonal representation.*

14 **Proof 1** *(i) We prove by contradiction. Assume that there exists such $\tilde{\rho}_N$ that satisfies Eqn. (9) when*
15 *$N$ is even. In the real domain, the irreps of $SO(2)$ are*

$$\begin{aligned}
\varphi_0^{SO(2)}(\theta) &= 1, \\
\varphi_k^{SO(2)}(\theta) &= \begin{bmatrix} \cos(k\tilde{\theta}) & -\sin(k\tilde{\theta}) \\ \sin(k\tilde{\theta}) & \cos(k\tilde{\theta}) \end{bmatrix}, \\
\theta &\in SO(2), \ k \in \mathbb{N}^*.
\end{aligned} \tag{23}$$

16 *Every representation of $SO(2)$ can be decomposed into the direct sum of the irreps in Eqn. (23) [6],*
17 *where each irrep may appear 0 or multiple times, and the direct sum $\oplus$ is matrix concatenation along*
18 *the diagonal, i.e.,*

$$A \oplus B = \begin{bmatrix} A & \\ & B \end{bmatrix}. \tag{24}$$

19   *As a special case, the decomposition of $\tilde{\rho}_N$ takes the following form: $\forall \theta \in SO(2)$,*

$$\tilde{\rho}_N(\theta) = A' \begin{bmatrix} \varphi_{i_1}^{SO(2)}(\theta) & & & \\ & \varphi_{i_2}^{SO(2)}(\theta) & & \\ & & \ddots & \\ & & & \varphi_{i_j}^{SO(2)}(\theta) \end{bmatrix} (A')^{-1}, \tag{25}$$

20   *where $A' \in GL(n, \mathbb{R})$, and $i_1, \cdots, i_j$ are non-negative integers.*

21   *The decomposition Eqn. (25) takes its form for all $\theta \in SO(2)$, obviously also holds for $\theta \in C_N$.*
22   *According to Eqn. (9), we have: $\forall \theta \in C_N$,*

$$\rho_{reg}^{C_N}(\theta) = A' \begin{bmatrix} \varphi_{i_1}^{SO(2)}(\theta) & & & \\ & \varphi_{i_2}^{SO(2)}(\theta) & & \\ & & \ddots & \\ & & & \varphi_{i_j}^{SO(2)}(\theta) \end{bmatrix} (A')^{-1}, \tag{26}$$

23   *Also, when $N$ is even, the decomposition of $\rho_{reg}^{C_N}$ is as follows: $\forall \theta \in C_N$,*

$$\rho_{reg}^{C_N}(\theta) = A \begin{bmatrix} \varphi_0^{C_N}(\theta) & & & & \\ & \varphi_1^{C_N}(\theta) & & & \\ & & \ddots & & \\ & & & \varphi_{\frac{N}{2}-1}^{C_N}(\theta) & \\ & & & & \varphi_{\frac{N}{2}}^{C_N}(\theta) \end{bmatrix} A^{-1}, \tag{27}$$

24   *where*

$$\begin{aligned} \varphi_0^{C_N}(\theta) &= 1, \\ \varphi_k^{C_N}(\theta) &= \begin{bmatrix} \cos(k\tilde{\theta}) & -\sin(k\tilde{\theta}) \\ \sin(k\tilde{\theta}) & \cos(k\tilde{\theta}) \end{bmatrix}, \\ \varphi_{\frac{N}{2}}^{C_N}(\theta) &= \cos(\tfrac{N}{2}\tilde{\theta}), \\ \theta &\in C_N, \ k \in \{1, 2, \cdots, \tfrac{N}{2} - 1\}, \end{aligned} \tag{28}$$

25   *and $A \in GL(n, \mathbb{R})$. When the irreps in the centering block diagonal matrix of the decomposition are*
26   *permuted in fixed order, such as the one in Eqn. (27) whose permutation is $\varphi_0^{C_N}(\theta), \cdots, \varphi_{N/2}^{C_N}(\theta)$,*
27   *the decomposition of $\rho_{reg}^{C_N}(\theta)$ is unique [6]. So it is necessary that the irreps in Eqn. (26) permute the*
28   *irreps in Eqn. (27).*

29   *However, when $N$ is even, $\rho_{reg}^{C_N}$ includes an additional irrep of $C_N$ than the case where $N$ is odd,*
30   *i.e., $\varphi_{N/2}^{C_N} = \cos(\tfrac{N}{2}\tilde{\theta})$, which cannot be expressed by any irreps in Eqn. (23). This results in*
31   *contradiction.*

32   ***(ii)** In Section 4.2 we have constructed a representation $\tilde{\rho}_N$ satisfying Eqn. (9). Here, we will prove*
33   *its uniqueness. For better illustration, we slightly modify the notations of Eqn. (14). As is shown in*
34   *(i), $\tilde{\rho}_N$ must take the following form: $\forall \theta \in SO(2)$,*

$$\tilde{\rho}_N(\theta) = A_1 \begin{bmatrix} \varphi_0^{SO(2)}(\theta) & & & \\ & \varphi_1^{SO(2)}(\theta) & & \\ & & \ddots & \\ & & & \varphi_{\frac{N-1}{2}}^{SO(2)}(\theta) \end{bmatrix} A_1^{-1}, \tag{29}$$

 *where*

$$\varphi_0^{SO(2)}(\theta) = 1,$$

$$\varphi_k^{SO(2)}(\theta) = \begin{bmatrix} \cos(k\tilde{\theta}) & -\sin(k\tilde{\theta}) \\ \sin(k\tilde{\theta}) & \cos(k\tilde{\theta}) \end{bmatrix}, \tag{30}$$

$$\theta \in SO(2),\ k \in \{1, 2, \cdots, \tfrac{N-1}{2}\},$$

*and $A_1 \in GL(n, \mathbb{R})$. Assume that there exists another $\overline{\rho}$ satisfying Eqn. (9). It is necessary that $\overline{\rho}$ shares the irreps of $\tilde{\rho}_N$, or else Eqn. (9) fails to hold for all $\theta \in C_N$. So $\overline{\rho}$ must take the following form: $\forall \theta \in SO(2)$,*

$$\overline{\rho}(\theta) = A_2 \begin{bmatrix} \varphi_0^{SO(2)}(\theta) & & & \\ & \varphi_1^{SO(2)}(\theta) & & \\ & & \ddots & \\ & & & \varphi_{\frac{N-1}{2}}^{SO(2)}(\theta) \end{bmatrix} A_2^{-1}, \tag{31}$$

*where $A_2 \in GL(n, \mathbb{R})$. As $\tilde{\rho}_N(\theta) = \overline{\rho}(\theta)$ for $\theta \in C_N$, from the equivalence of the right hand sides of Eqn. (29) and Eqn. (31), we have that for $\forall \theta \in C_N$,*

$$A_2^{-1}A_1 \begin{bmatrix} \varphi_0^{SO(2)}(\theta) & & & \\ & \varphi_1^{SO(2)}(\theta) & & \\ & & \ddots & \\ & & & \varphi_{\frac{N-1}{2}}^{SO(2)}(\theta) \end{bmatrix} = \begin{bmatrix} \varphi_0^{SO(2)}(\theta) & & & \\ & \varphi_1^{SO(2)}(\theta) & & \\ & & \ddots & \\ & & & \varphi_{\frac{N-1}{2}}^{SO(2)}(\theta) \end{bmatrix} A_2^{-1}A_1. \tag{32}$$

*The matrix $\varphi_0^{SO(2)}(\theta) \oplus \varphi_1^{SO(2)}(\theta) \oplus \cdots \oplus \varphi_{\frac{N-1}{2}}^{SO(2)}(\theta)$ is block diagonal with each block $\varphi_i^{SO(2)}(\theta)$, $i = 0, 1, \cdots, (N-1)/2$. Now, we partition the matrix $A_2^{-1}A_1$ into a block matrix by exactly the same way that the block diagonal matrix is partitioned. We use the notation $(A_2^{-1}A_1)_{ij}$ to represent the block in the $i^{th}$ row and $j^{th}$ column, then for $\forall \theta \in C_N$,*

$$(A_2^{-1}A_1)_{ij}\varphi_j^{SO(2)}(\theta) = \varphi_i^{SO(2)}(\theta)(A_2^{-1}A_1)_{ij}, \tag{33}$$

$$\Leftrightarrow (A_2^{-1}A_1)_{ij}\varphi_j^{C_N}(\theta) = \varphi_i^{C_N}(\theta)(A_2^{-1}A_1)_{ij} \tag{34}$$

*where $i = 0, \cdots, (N-1)/2, j = 0, \cdots, (N-1)/2$. According to Schur's Lemma [6], when $i = j = 0$, we have $(A_2^{-1}A_1)_{ij} = r_0$, where $r_0 \in \mathbb{R}$. When $i = j \neq 0$, we have $(A_2^{-1}A_1)_{ij} = r_i R_i$, where $r_i \in \mathbb{R}$ and $R_i \in SO(2)$. Otherwise, we have $(A_2^{-1}A_1)_{ij} = O$, where $O$ is the zero matrix. Now we can represent $A_1$ with $A_2$:*

$$A_1 = A_2 \begin{bmatrix} r_0 & & & \\ & r_1 R_1 & & \\ & & \ddots & \\ & & & r_{\frac{N-1}{2}} R_{\frac{N-1}{2}} \end{bmatrix}. \tag{35}$$

*Plugging Eqn. (35) into Eqn. (29), we get that for any $\theta \in SO(2)$,*

$$\tilde{\rho}_N(\theta) = A_2 \begin{bmatrix} r_0 & & & \\ & r_1 R_1 & & \\ & & \ddots & \\ & & & r_{\frac{N-1}{2}} R_{\frac{N-1}{2}} \end{bmatrix} \begin{bmatrix} \varphi_0^{SO(2)}(\theta) & & & \\ & \varphi_1^{SO(2)}(\theta) & & \\ & & \ddots & \\ & & & \varphi_{\frac{N-1}{2}}^{SO(2)}(\theta) \end{bmatrix} \begin{bmatrix} r_0 & & & \\ & r_1 R_1 & & \\ & & \ddots & \\ & & & r_{\frac{N-1}{2}} R_{\frac{N-1}{2}} \end{bmatrix}^{-1} A_2^{-1}. \tag{36}$$

*For $i = 1, \cdots, (N-1)/2$, the matrices $R_i$ and $\varphi_i^{SO(2)}(\theta)$ commute since they are all rotation matrices, so $r_i R_i \varphi_i^{SO(2)}(\theta) R_i^{-1} r_i^{-1} = \varphi_i^{SO(2)}(\theta)$. So $\tilde{\rho}_N(\theta) = \overline{\rho}(\theta)$ for $\theta \in SO(2)$, proving the uniqueness of $\tilde{\rho}_N$.*

53 *(iii) From Eqn. (29), we can get that*

$$
A_1^\top \tilde{\rho}_N(\theta)^\top \tilde{\rho}_N(\theta) A_1 = \begin{bmatrix} \varphi_0^{SO(2)}(\theta) & & & \\ & \varphi_1^{SO(2)}(\theta) & & \\ & & \ddots & \\ & & & \varphi_{\frac{N-1}{2}}^{SO(2)}(\theta) \end{bmatrix}^\top A_1^\top A_1 \begin{bmatrix} \varphi_0^{SO(2)}(\theta) & & & \\ & \varphi_1^{SO(2)}(\theta) & & \\ & & \ddots & \\ & & & \varphi_{\frac{N-1}{2}}^{SO(2)}(\theta) \end{bmatrix}.
$$
(37)

54 *As $\rho_{reg}^{C_N}$ is orthogonal representation, Eqn. (9) tells us that $\tilde{\rho}_N(\theta)^\top \tilde{\rho}_N(\theta) = I$ for $\theta \in C_N$. Note that*
55 *all the irreps are orthogonal representations, we have*

$$
A_1^\top A_1 \begin{bmatrix} \varphi_0^{SO(2)}(\theta) & & & \\ & \varphi_1^{SO(2)}(\theta) & & \\ & & \ddots & \\ & & & \varphi_{\frac{N-1}{2}}^{SO(2)}(\theta) \end{bmatrix} = \begin{bmatrix} \varphi_0^{SO(2)}(\theta) & & & \\ & \varphi_1^{SO(2)}(\theta) & & \\ & & \ddots & \\ & & & \varphi_{\frac{N-1}{2}}^{SO(2)}(\theta) \end{bmatrix} A_1^\top A_1,
$$
(38)

56 *for $\theta \in C_N$. Partition the matrix $A_1^\top A_1$ into a block matrix by exactly the same way that*
57 *the block diagonal matrix is partitioned, then Eqn. (38) is equivalent to, for all $\theta \in C_N$,*
58 *$\varphi_i^{SO(2)}(\theta)(A_1^\top A_1)_{ij} = (A_1^\top A_1)_{ij}\psi_j^{SO(2)}(\theta) \Leftrightarrow \varphi_i^{C_N}(\theta)(A_1^\top A_1)_{ij} = (A_1^\top A_1)_{ij}\psi_j^{C_N}(\theta)$, where*
59 *$i = 0, \cdots, (N-1)/2, j = 0, \cdots, (N-1)/2$. According to Schur's Lemma, when $i = j = 0$, we*
60 *have $(A_1^\top A_1)_{ij} = r_0', r_0' \in \mathbb{R}$. When $i = j \neq 0$, we have $(A_1^\top A_1)_{ij} = r_i' R_i'$, where $r_i' \in \mathbb{R}$ and*
61 *$R_i' \in SO(2)$. Otherwise, $(A_1^\top A_1)_{ij} = O$. So it is obvious that $A_1^\top A_1$ commutes with the block*
62 *diagonal matrix $\varphi_0^{SO(2)}(\theta) \oplus \varphi_1^{SO(2)}(\theta) \oplus \cdots \oplus \varphi_{\frac{N-1}{2}}^{SO(2)}(\theta), \forall \theta \in SO(2)$. Then, for any $\theta \in SO(2)$,*

$$
\begin{aligned}
\tilde{\rho}_N(\theta)^\top \tilde{\rho}_N(\theta) =& (A_1^\top)^{-1} \begin{bmatrix} \varphi_0^{SO(2)}(\theta) & & & \\ & \varphi_1^{SO(2)}(\theta) & & \\ & & \ddots & \\ & & & \varphi_{\frac{N-1}{2}}^{SO(2)}(\theta) \end{bmatrix}^\top A_1^\top A_1 \begin{bmatrix} \varphi_0^{SO(2)}(\theta) & & & \\ & \varphi_1^{SO(2)}(\theta) & & \\ & & \ddots & \\ & & & \varphi_{\frac{N-1}{2}}^{SO(2)}(\theta) \end{bmatrix} A_1^{-1} \\
=& (A_1^\top)^{-1} \begin{bmatrix} \varphi_0^{SO(2)}(\theta) & & & \\ & \varphi_1^{SO(2)}(\theta) & & \\ & & \ddots & \\ & & & \varphi_{\frac{N-1}{2}}^{SO(2)}(\theta) \end{bmatrix}^\top \begin{bmatrix} \varphi_0^{SO(2)}(\theta) & & & \\ & \varphi_1^{SO(2)}(\theta) & & \\ & & \ddots & \\ & & & \varphi_{\frac{N-1}{2}}^{SO(2)}(\theta) \end{bmatrix} A_1^\top A_1 A_1^{-1} \\
=& (A_1^\top)^{-1} A_1^\top A_1 A_1^{-1} = I,
\end{aligned}
$$
(39)

63 *which completes the proof.*

## B.2  Proof of Theorem 2

65 **Theorem 2** *Assume a GET $\psi$, whose types of input, intermediate, and output feature fields are $\rho_{local}$,*
66 *$k_i \rho_{reg}^{C_N}$ and $\rho_0$, respectively, where $k_i$ is the number of regular fields in the $i^{th}$ intermediate feature*
67 *field. Denote $f$ as the input feature field on triangle mesh $M$, and assume that the norm of the feature*
68 *map $\|f_w\|$ is bounded by a constant $C$. Gauges $w$ and $w'$ are linked by transformation $g$. Further*
69 *suppose that $\psi$ is Lipschitz continuous with constant $L$, then we have:*

70 *(i) If $g_p \in C_N$ for every mesh vertex $p \in M$, then $\psi(f_w) = \psi(f_{w'})$.*

71 *(ii) For general $g_p \in SO(2)$, we have $\|\psi(f_w) - \psi(f_{w'})\| \leq \frac{\pi L}{N} C$.*

72 **Proof 2** *(i) Since the equivariance of the multi-head self-attention (Eqn. (5)) directly follows from*
73 *the equivariance of single-head self-attention (Eqn. (21)), we only give the equivariance proof of*
74 *Eqn. (21) here. For simplicity, we omit the head $h$ in this proof.*

*Firstly, we show the gauge invariance of attention score in Eqn. (22) by showing that the score function is gauge invariant. We use $S_w$ to denote the score function under the gauge $w$, and use $S_{w'}$ under the gauge $w'$. As the feature fields of the intermediate layers are regular fields whose representation are permutation matrices for gauge transformations in $C_N$, composing the element-wise ReLU preserves gauge equivariance. Eqn. (40) holds for all intermediate layers when $g_p \in C_N$:*

$$ReLU(\rho(g_p)f_w(g)) = \rho(g_p)ReLU(f_w(g)). \tag{40}$$

*As is introduced in Section 3, the quantities in different gauges are related as follows:*

$$w'_p = w_p g_p, \tag{41}$$

$$f_{w'}(q) = \rho_{in}(g_q^{-1})f_w(q), \tag{42}$$

$$g_{q \to p}^{w'} = g_p^{-1} g_{q \to p}^{w} g_q, \tag{43}$$

$$u'_q = g_p^{-1} u_q. \tag{44}$$

*Using the key and query function in Section 4.4, we have*

$$S_w = P(ReLU(W_K f_w(p) + W_Q \rho_{in}(g_{q \to p}^{w})f_w(q))). \tag{45}$$

*Under the gauge $w'$, it is*

$$\begin{aligned} S_{w'} =&P(ReLU(W_K f_{w'}(p) + W_Q \rho_{in}(g_{q \to p}^{w'})f_{w'}(q))) & \text{(46)}\\ =&P(ReLU(W_K \rho_{in}(g_p^{-1})f_w(p) + W_Q \rho_{in}(g_p^{-1} g_{q \to p}^{w} g_q)\rho_{in}(g_q^{-1})f_w(q))) & \text{(47)}\\ =&P(ReLU(\rho_{out}(g_p^{-1})W_K f_w(p) + \rho_{out}(g_p^{-1})W_Q \rho_{in}(g_{q \to p}^{w})f_w(q))) & \text{(48)}\\ =&P(\rho_{out}(g_p^{-1})ReLU(W_K f_w(p) + \rho_{out}(g_p^{-1})W_Q \rho_{in}(g_{q \to p}^{w})f_w(q))) & \text{(49)}\\ =&P(ReLU(W_K f_w(p) + W_Q \rho_{in}(g_{q \to p}^{w})f_w(q))), & \text{(50)} \end{aligned}$$

*where Eqn. (46) to Eqn. (47) is according to relationship of quantities in different gauges, Eqn. (47) to Eqn. (48) is using the property that $W_K$ and $W_Q$ satisfy Eqn. (17a), Eqn. (48) to Eqn. (49) is from Eqn. (40), and Eqn. (49) to Eqn. (50) is based on the fact that the output of average pooling stays the same under any permutation of the components.*

*Now we show the gauge equivariance of the value function. Under the gauge $w$, the value function is*

$$V_{u_q}(f'_w(q)) = W_V(u_q)\rho_{in}(g_{q \to p}^{w})f_w(q), \tag{51}$$

*under the gauge $w'$, it is*

$$V_{u'_q}(f'_{w'}(q)) = W_V(u'_q)\rho_{in}(g_{q \to p}^{w'})f_{w'}(q), \tag{52}$$

*Plugging equations (41)–(44) into Eqn. (52), we get*

$$\begin{aligned} V_{u'_q}(f'_{w'}(q)) =&W_V(g_p^{-1}u_q)\rho_{in}(g_p^{-1} g_{q \to p}^{w} g_q)\rho_{in}(g_q^{-1})f_w(q) & \text{(53)}\\ =&\rho_{out}(g_p^{-1})W_V(u_q)\rho_{in}(g_p)\rho_{in}(g_p^{-1} g_{q \to p}^{w} g_q)\rho_{in}(g_q^{-1})f_w(q) & \text{(54)}\\ =&\rho_{out}(g_p^{-1})W_V(u_q)\rho_{in}(g_{q \to p}^{w})f_w(q) & \text{(55)}\\ =&\rho_{out}(g_p^{-1})V_{u_q}(f'_w(q)). & \text{(56)} \end{aligned}$$

*So the single-head attention Eqn. (21) is exactly equivariant to gauge transformations in $C_N$. Also, the stack of gauge equivariant layers is gauge equivariant, hence $\psi$ is gauge equivariant. According to the type of its input and output feature fields, we have $\psi(f_w) = \psi(f_{w'})$.*

*(ii) For any gauge transformation $g_p$, there exists $\overline{g}_p \in C_N$ such that the rotation angle $\tilde{\theta}_p$ with respect to $g_p^{-1}\overline{g}_p$ lies in $[-\frac{\pi}{N}, \frac{\pi}{N}]$. Express the manifold equation as $\overline{w} = w \cdot \overline{g}$, then we have $\psi(f_w) = \psi(f_{\overline{w}})$, as is shown by (i). Note that the norm of a feature map here is defined as the Euclidean norm of a zipped vector produced by aligning the feature vectors of all points on the mesh*

*into one column. Then we have*

$$\|\psi(f_w) - \psi(f_{w'})\| = \|\psi(f_{\overline{w}}) - \psi(f_{w'})\| \tag{57}$$

$$\leq L\|f_{\overline{w}} - f_{w'}\| \tag{58}$$

$$= L\Big(\sum_p \|f_{\overline{w}}(p) - f_{w'}(p)\|^2\Big)^{\frac{1}{2}} \tag{59}$$

$$= L\Big(\sum_p \|(I - \rho_{local}(g_p^{-1}\overline{g}_p))f_{\overline{w}}(p)\|^2\Big)^{\frac{1}{2}} \tag{60}$$

$$\leq L\Big(\sum_p \|I - \rho_{local}(g_p^{-1}\overline{g}_p)\|_2^2\|f_{\overline{w}}(p)\|^2\Big)^{\frac{1}{2}}, \tag{61}$$

*where $\|\cdot\|_2$ is the matrix spectral norm, and*

$$\|I - \rho_{local}(g_p^{-1}\overline{g}_p)\|_2 = \left\|\begin{bmatrix} 1 - \cos\tilde{\theta}_p & \sin\tilde{\theta}_p & 0 \\ -\sin\tilde{\theta}_p & 1 - \cos\tilde{\theta}_p & 0 \\ 0 & 0 & 0 \end{bmatrix}\right\|_2 = 2\left|\sin(\frac{\tilde{\theta}_p}{2})\right| \leq |\tilde{\theta}_p| \leq \frac{\pi}{N}. \tag{62}$$

*So*

$$\|\psi(f_w) - \psi(f_{w'})\| \leq L\Big(\sum_p (\frac{\pi}{N}\|f_{\overline{w}}(p)\|)^2\Big)^{\frac{1}{2}} = \frac{\pi L}{N}\|f_{\overline{w}}\| \leq \frac{\pi L}{N}C. \tag{63}$$

## C   Solution of Equivariant Constraint

Here, we provide the detailed process of computing solution basis of Eqn. (15) for all $\Theta \in C_N$. Firstly, we show that Eqn. (15) holds for all $\Theta \in C_N$ is equivalent to it holds for one matrix $\Theta_0$ with the corresponding rotation angle $\theta_0 = 2\pi/N$, *i.e.*,

$$\Theta_0 = \begin{bmatrix} \cos\dfrac{2\pi}{N} & -\sin\dfrac{2\pi}{N} \\ \sin\dfrac{2\pi}{N} & \cos\dfrac{2\pi}{N} \end{bmatrix}. \tag{64}$$

The sufficiency is obvious, here we only show the necessity. In Section 4.3, we use Taylor expansion Eqn. (16) to solve the equivariance constraint Eqn. (15). The Taylor coefficients $\{W_0, W_1, \cdots\}$ solve equations (17) if and only if $W_V(u) = W_0 + W_1 u_1 + W_2 u_2 + \cdots$ solves Eqn. (15). Known that $\{W_0, W_1, \cdots\}$ solve equations (17) for $\Theta_0$, then Eqn. (15) holds for this $\Theta_0$, i.e.,

$$W_V(\Theta_0^{-1}u) = \rho_{out}(\Theta_0^{-1})W_V(u)\rho_{in}(\Theta_0). \tag{65}$$

Now we prove by induction that $W_V(u)$ solves Eqn. (15) for $\Theta_0^k$ for any $k \in \mathbb{N}^*$, where $\Theta_0^k \in C_N$ is the rotation matrix with respect to angle $k\theta_0$, i.e.,

$$\Theta_0^n = \begin{bmatrix} \cos k\dfrac{2\pi}{N} & -\sin k\dfrac{2\pi}{N} \\ \sin k\dfrac{2\pi}{N} & \cos k\dfrac{2\pi}{N} \end{bmatrix}. \tag{66}$$

One can easily verify the correctness of Eqn. (66) by Eqn. (65).

Eqn. (65) is the statement when $k = 1$. Suppose that it holds for $k = l$, where $l \in \mathbb{N}^*$, i.e.,

$$W_V((\Theta_0^l)^{-1}u) = \rho_{out}((\Theta_0^l)^{-1})W_V(u)\rho_{in}(\Theta_0^l). \tag{67}$$

When $k = l + 1$, one can derive that

$$W_V((\Theta_0^{l+1})^{-1}u) = W_V((\Theta_0^{-1}(\Theta_0^l)^{-1}u) \tag{68}$$

$$= \rho_{out}(\Theta_0^{-1})W_V((\Theta_0^l)^{-1}u))\rho_{in}(\Theta_0) \tag{69}$$

$$= \rho_{out}(\Theta_0^{-1})\rho_{out}((\Theta_0^l)^{-1})W_V(u)\rho_{in}(\Theta_0^l)\rho_{in}(\Theta_0) \tag{70}$$

$$= \rho_{out}((\Theta_0^{l+1})^{-1})W_V(u)\rho_{in}(\Theta_0^{l+1}), \tag{71}$$

which suggests that the statement still holds. So Eqn. (72) holds for every $k \in \mathbb{N}^*$:

$$W_V((\Theta_0^k)^{-1}u) = \rho_{out}((\Theta_0^k)^{-1})W_V(u)\rho_{in}(\Theta_0^k), \tag{72}$$

which proves the necessity. So we only have to solve the constraint Eqn. (15) for $\Theta_0$. More general, for any group, we only need to solve the constraint Eqn. (15) for one set of generators of the group.

As is shown in Section 4.3, we can solve the linear equations in (17) with the same order independently. Now consider the equations in (17) with order $n$. For convenience, denote the matrices $B_0, B_1, \cdots, B_n$ are the coefficients of the terms $u_1^n, u_1^{n-1}u_2, \cdots, u_2^n$, respectively. The relationship with the coefficients in Eqn. (16) is that $B_i = W_{(n+1)n/2+i}$. Then the equations in (17) with order $n$ can be rewritten as

$$\sum_{j=0}^{n} F_{ij}B_j = \rho_{out}(\Theta_0^{-1})B_i\rho_{in}(\Theta_0), \text{ for } i = 0, 1, \cdots, n, \tag{73}$$

where $F \in \mathbb{R}^{(n+1)\times(n+1)}$ is a matrix. For example, when the order $n = 1$, $F = \Theta_0$. To simplify computation, we stretch the matrices $B_0, B_1, \cdots, B_n$ and align them into a long $((n+1) \times C_{out} \times C_{in})$-dimensional vector $\tilde{B}$, i.e.

$$\tilde{B}_{i\times C_{out}\times C_{in}+j\times C_{in}+k} = (B_i)_{jk}. \tag{74}$$

Then the equation (73) is equivalent to: $\forall i, t, l, s.t., 0 \le i \le n, 1 \le t \le C_{out}, 1 \le l \le C_{in}$,

$$\sum_{j} F_{ij}(B_j)_{tl} = \sum_{t',l'} \rho_{out}(\Theta_0^{-1})_{tt'}(B_i)_{t'l'}\rho_{in}(\Theta_0)_{l'l} \tag{75}$$

$$\Longleftrightarrow \sum_{j,t',l'} F_{ij}\delta_{tt'}\delta_{ll'}(B_j)_{t'l'} = \sum_{j,t',l'} \delta_{ij}\rho_{out}(\Theta_0^{-1})_{tt'}\rho_{in}(\Theta_0)_{ll'}^{\top}(B_j)_{t'l'} \tag{76}$$

According to the definition of the Kronecker product $\otimes$

$$\Longleftrightarrow F \otimes I_{C_{out}} \otimes I_{C_{in}}\tilde{B} = (I_{n+1} \otimes (\rho_{out}(\Theta_0^{-1}) \otimes \rho_{in}^{\top}(\Theta_0)))\tilde{B}. \tag{77}$$

Then the equation (73) can be reduced to a more compact linear equation:

$$(I_{n+1} \otimes (\rho_{out}(\Theta_0^{-1}) \otimes \rho_{in}^{\top}(\Theta_0)) - F \otimes I_{C_{out}} \otimes I_{C_{in}})\tilde{B} = 0. \tag{78}$$

where $I_{C_{out}}$, $I_{C_{out}}$ and $I_{n+1}$ are the identity matrices of dimension $C_{out}$, $C_{in}$ and $n+1$, respectively. The solution bases of Eqn. (78) can be efficiently computed via SVD.

# D  Experiment Details

Before going into the experiments, we introduce several structures adopted in our neural networks. All experiments are carried on Ubuntu 20.04 machine with NVIDIA RTX 3090 GPU.

**Linear Layer.** The linear layer receives an input and produces an output that is the linear transformation of the input. Since our network is gauge equivariant, the linear transformation matrix has to satisfy the Eqn. (17a).

**Average Pooling.** Wiersma et al. [9] propose an average pooling method we use here. Firstly, the Farthest Point Sampling algorithm [2] is employed to sample the representative points, giving out the vertices of the pooled mesh. Then every non-sampled point in the original mesh is clustered into its geodesically nearest representative point among all representative points. At last, the feature vector of each representative point in the pooled mesh is taken as the average of all the feature vectors of its cluster:

$$\overline{f}_w(p) = \frac{1}{|C_p|} \sum_{q \in C_p} \rho_{in}(g_{q \to p}^w)f_w(q), \tag{79}$$

where $C_p$ is the cluster of $p$, and $\overline{f}_w(p)$ is the value of pooled feature vector.

To clarify, the Average Pooling used in supplementary materials refers to the pooling method proposed in [9] with respect to mesh vertices, different from the average pooling operation in computing attention score (in Section 4.4)

**Global Average Pooling.** The Global Average Pooling layer takes the average of every component of the feature vectors on all vertices of the mesh, producing a global feature vector.

**Group Pooling.** For each component of the feature vector in the regular field under a specified gauge, the Group Pooling layer [8] outputs its maximum element, producing a gauge invariant scalar field.

**Unpooling.** The Unpooling layer is like the inverse of the average pooling layer. It upsamples the feature map by parallel transporting the feature vector from the representative point to each point in the original cluster.

## D.1 Data Preprocessing

The datasets used in this paper are all in the form of triangle meshes. Given the mesh data of a sample, we compute its surface area by summing up the areas of all faces, and then scale it into 1. For each point $p$, we construct the neighborhoods $\mathcal{N}_p$ in Eqn. (22) by selecting all vertices within geodesic distance $\sigma$ to $p$. Then, the mesh data can be processed into a graph where the edge connection represents neighborhood relationship. For each vertex and its neighbor vertices, we use the Vector Heat Method [7] to precompute the logarithmic map and the rotation angle induced by parallel transport from each neighbor vertex to the center.

After the downsampling of the pooling layer and neighborhood reconstruction, one can obtain a smaller scale graph whose vertices are a subset of the vertices in the original mesh. Following [9], we incorporate graph structures in different scales into a multi-scale graph. Then the logarithmic map and parallel transport can be computed in one pass. Our model receives pointwise local coordinate input (*i.e.* $X$ in Section 4.5) to guarantee $SO(3)$ invariance, which can also be computed in advance.

## D.2 SHREC Classification

The neural network used in the shape classification task is lightweight but successful. Input features in Section 4.5 are first processed by a linear layer, producing a feature field of type $12\rho_{reg}^{C_N}$. After that, a single ResNet block [3] is used, with the radius $\sigma$ set to $0.2$, *i.e.*, we take into account all the vertices within a geodesic distance of $0.2$ as the neighbors $\mathcal{N}_p$ in Eqn. (22). The output of the ResNet block is also a $12\rho_{reg}^{C_N}$ feature field. The followings are a group pooling layer and a global average pooling layer. At last, a fully connected layer is attached and the softmax function outputs the final probabilities of each class. The architecture is visualized in Figure 5. The network is trained for 70 epochs using the Adam optimizer [4] with an initial learning rate of $0.005$ and is divided by 10 at $41^{th}$ epoch. The order of the cyclic group $C_N$ is set to 9. To leverage robustness, every input mesh is scaled with a factor of random variable uniformly distributed in $[0.85, 1.15]$ in training.

## D.3 Human Body Segmentation

Following [9], to reduce training time, we use Farthest Point Sampling algorithm to select $1024$ vertices from the original mesh data in training and testing. U-ResNet is a prominent architecture in the field of geometric deep learning [1]. It has a multi-scale structure with several pooling and unpooling layers. Here we employ the method in [9] for adapting these layers to mesh data. Our models have two scales and the neighborhood radii are $0.2$ and $0.4$, respectively. We use four ResNet blocks in each stage of feature transformation. Again we set $N = 9$ here, so all the feature fields in intermediate layers are regular fields of $C_9$. The architecture is visualized in Figure 6. We train the network for $50$ epochs with the Adam algorithm. The learning rate is initialized as $0.01$ and is divided by 10 at $31^{th}$ epoch, and further divided into half at $41^{th}$ epoch.

## D.4 Ablation Study

**Local Coordinate.** In Section 4.5, we have proposed to incorporate local coordinates to make our model rotation invariant. To verify their superiority, we adopt a baseline model whose inputs are raw $xyz$ coordinates. Like RGB channels in color images, the $xyz$ coordinates are treated as three scalar fields, $3\rho_0$. For a fair comparison, the baseline model is identical to our state-of-the-art model except for the first layer.

The comparison is carried out in three settings: No rotations on the training dataset and no rotations on the test dataset (N/N), no rotations on the training dataset and rotate on the test dataset (N/R), and

rotate on the training dataset and rotate on the test dataset (R/R). As is shown in Table 5, applying the $\rho_{local}$ feature field consistently improves model accuracy in all cases as it enables our model to be invariant to $SO(3)$ rotations intrinsically.

Table 5: Model accuracy in the human body segmentation task with respect to different types of inputs.

| Input Type | (N/N) | (N/R) | (R/R) |
|---|---|---|---|
| $3\rho_0$ | 91.5% | 90.9 % | 91.6% |
| $\rho_{local}$ (Ours) | **92.6%** | **92.6%** | **92.6%** |

**Parallel Transport Methods.** Parallel transport carries the information of surface geometry, playing a crucial role in assuring gauge equivariance. Here we replace our parallel transport method with two baseline methods, truncation [10] and interpolation [5], to validate the effectiveness of our method. The results are shown in Table 6.

All the models listed in Table 6 only differ in parallel transport methods. The None setting serves as the control group where parallel transport is not used. Its setup is for showing the effectiveness of parallel transport. Our model shows conspicuous superiority to all baselines. Compared with ours, parallel transport with interpolation fails to preserve the norm of feature vector while truncation disregards the relative orientation information to some extent.

Table 6: Model accuracy in the human body segmentation task with respect to different parallel transport methods.

| Method | Ours | Interpolation | Truncation | None |
|---|---|---|---|---|
| Accuracy(%) | **92.6** | 92.1 | 91.3 | 86.7 |

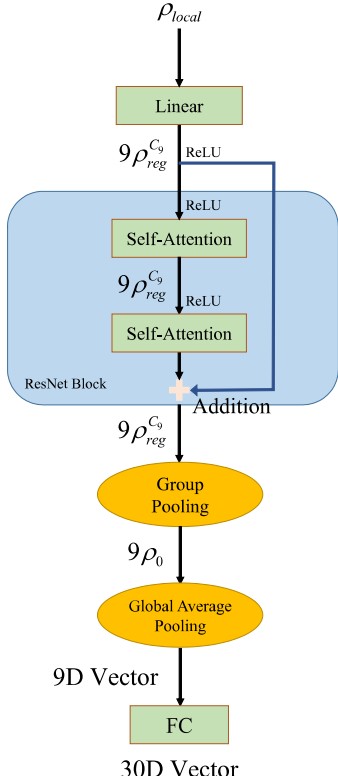

Figure 5: The state-of-the-art neural network architecture for shape classification task.

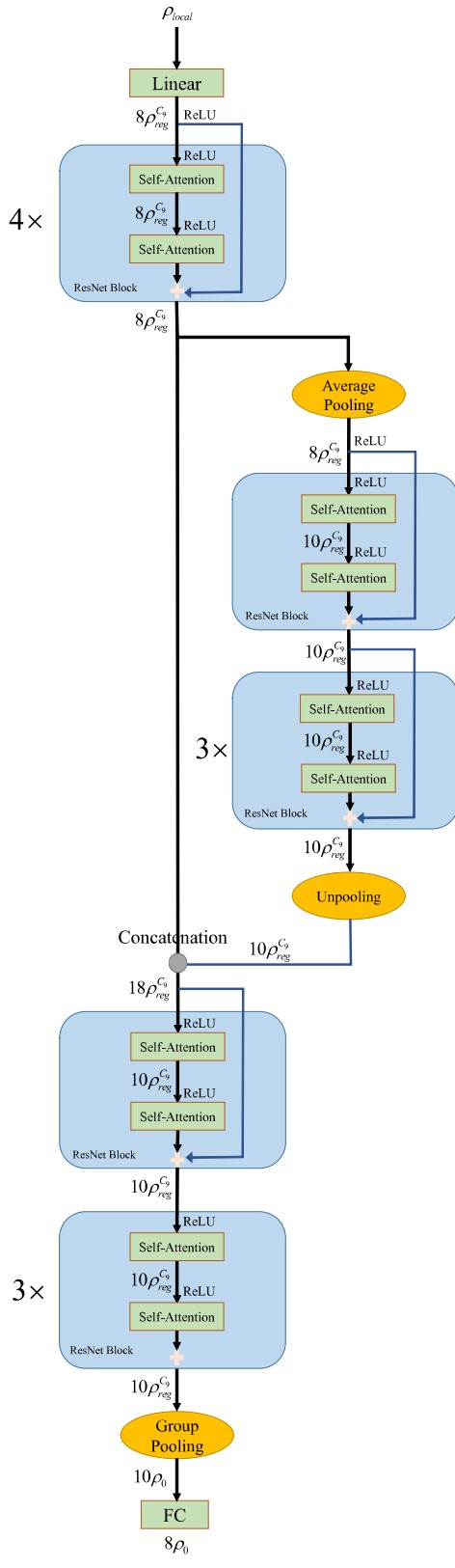

Figure 6: The state-of-the-art neural network architecture for shape segmentation task.