# OpenReview forum: "Gauge Equivariant Transformer"
_NeurIPS.cc/2021/Conference — NeurIPS 2021 Poster_

### Official Review · Reviewer_3RJ5 · 2021-07-16

**Rating:** 6
**Confidence:** 3

**Summary:**

The paper presents a gauge equivariant transformer for geometric learning. The gauge equivariance aims to solve the orientation ambiguity when doing the convolution or local aggregation operations within a neighborhood. Specifically, this work achieves the gauge equivariant self-attention within the neighborhood defined by mesh data. Different from existing gauge equivariant layers, the paper proposes to accommodate the parallel transport in the regular field. As a result, the gauge equivariance is up to a certain error bound. Further, this work validates the effectiveness on two datasets including shape classification (SHREC) and human body segmentation.

**Limitations And Societal Impact:**

From my viewpoint, I suggest discussing the limitation in terms of efficiency.

**Main Review:**

From my viewpoint, this work will have a certain impact -- incorporating the gauge equivariance into the recently popular transformer. The main concern I have is about the comparison with other gauge equivariant CNN, especially given that the paper shows the negligible contribution of attention to the performance improvement in ablation study.

Originality:
Gauge equivariant self-attention is novel. Although it’s a combination between existing gauge equivariance and self-attention, this work closes the gap in the transformer. In addition, the work presents a new way to accommodate parallel transport, which is interesting to me.

Quality:
The paper is of high quality in terms of theoretical analysis. But I’m a bit concerned about the experimental validation.
- Missing baselines related to gauge equivariance. For example, the geodesic CNN [25] also tried to solve the orientation ambiguity problem. Also, there is no other method for comparison in the tables that also incorporate gauge equivariance.
- I would appreciate it more if the paper could discuss and compare with gauge equivariant mesh CNN [46, 12].

Clarity:
Overall, the paper is well-written. But I still have several comments. Hope the author could address them during the rebuttal.
- What’s the motivation for using a regular field for the intermediate features?
- The gauge equivariance is not strict (up to error bound). I am not sure if the title will be misleading.
- What’s the efficiency of the proposed method?
- Adding citations into table 1 and 2 would be more readable.


Significance:
This paper injects the gauge equivariance into the recent popular transformer. However, due to the missing comparison with other gauge equivariant CNN, I am not sure if the result is significant. Further, the ablation study shows that the key to the improvement is due to the gauge equivariance. Does this mean that convolution, as used in [46, 12], would be better? I would like to hear more from the authors during the rebuttal.


### Post rebuttal
I have the feedback from the authors and other reviews. Since the feedback addressed my concerns, I would like to improve my rating.

**Time Spent Reviewing:**

6

---

> ### Author Response · Authors · 2021-08-10
> **Initial Response**
>
> We sincerely thank Reviewer 3RJ5 for appreciating the novelty of this paper and the theoretical parts.
>
> Now we will answer to all of your questions, suggestions, comments and concerns:
>
> ## Q1: About the contribution of attention to the performance improvement
>
> A1: Our GET achieves the state-of-the-art performance on the shape segmentation task (92.6%), 0.3% better than the non-attentive baseline. Now we find that it is not always the case, when applying GET model and the non-attentive baseline to the shape classification task, they give the accuracies of 99.2% and 96.7%, respectively. This yields a significant performance gain (2.5%), showing the great effect of self-attention.
>
> As for the shape segmentation task, the reason why the performance improvement is not fully realized here is perhaps relevant to the labels of the neighboring points. Since each point in the local neighborhood is labeled, and in some cases, the labels are of several types, the attention machanism may have no obvious advantage comparing with convolution then.
>
> We make the first step towards applying principled Transformer on the manifold, and we believe that the research progress with respect to Transformers on Vision task would help facilitate extending and improving our model.
>
> ## Q2: Baselines related to gauge equivariance
>
> A2: Actually, the HSN [46] compared in classification and segmentation tasks is gauge equivariant, with its accuracies shown in Table 1 and Table 2. Beside this, the works of "Gauge Equivariant Convolutional Networks and the Icosahedral CNN" [10] and "Gauge Equivariant Mesh CNN (GEM-CNN)" [12] are gauge equivariant, in which [12] is an extension of [10] on regular meshgrids. GEM-CNN gives 93.3% accuracy on the classification task, and gives 91.3% accuracy on the segmentation task, both are inferior to ours, which are 99.2% on the classification task and 92.6% on the segmentation task, respectively.
>
> Up to now, the number of the works related to gauge equivariance is limited. Many of the intrinsic methods, including Geodesic CNN [25], MoNet [27] and ACNN [4], are inherently gauge invariant as they directly select one specific gauge to solve the orientation ambiguity problem. Alternatively, one may say that there is no "variation" on the gauges.
>
> In addition, we report the accuracies of Geodesic CNN [25] on two tasks: 73.9% on the classification task, 86.4% on the segmentation task, both are much inferior to ours, which are 99.2% on the classification task and 92.6% on the segmentation task, respectively.
>
> ## Q3: Discussion and comparison with gauge equivariant CNNs
>
> A3: Gauge Equivariant Convolutional Networks and the Icosahedral CNN [10] is the seminal work of gauge equivariant networks, implemented on regular grids. Gauge Equivariant Mesh CNN (GEM-CNN) [12] and HSN [46] are the successful extensions of [10] on general meshgrids. It is recognized in [12] that the element wise non-linearities, such as ReLU, generally outperform norm non-linearities [46] and gated non-linearities, but element wise non-linearities are not gauge equivariant for features in [46] and [12]. Instead, [12] proposes a new activation function called RegularNonlinearity, but the Fourier transformation incorporated in it induces extra computational burden. Our GET incorporates regular fields in intermediate layers, enabling the usage of element wise non-linearities while preserving gauge equivariance.
>
> In addition, we provide the results of GEM-CNN and HSN in both two tasks. In the classification task, GEM-CNN gives 93.3% accuracy and HSN gives 96.1% accuracy; In the segmentation task, GEM-CNN gives 91.3% accuracy and HSN gives 91.1% accuracy. All of the results are inferior to ours, which are 99.2% on the classification task and 92.6% on the segmentation task, respectively.
>
> ## Q4: What’s the motivation for using a regular field for the intermediate features?
>
> A4: There are several reasons for using the regular field:
>
> (1) As indicated in the paper [44], regular field is generally the most powerful feature type of equivariant networks.
>
> (2) Taking in the regular field enables us to use the element wise non-linearity in the network which in general has a better performance than norm non-linearities and gated non-linearities which are adopted for feature fields of irreducible representations.
>
> ## Q5: What’s the efficiency of the proposed method?
>
> A5: The aspects of the efficiency of our GET include higher accuracy, fewer parameters and less computation time. As is shown in the main paper, compared to the second best baselines, GET achieves 3.1% higher accuracy with about 1/7 parameters on the classification task, and achieves 0.3% higher accuracy with about 1/15 parameters on the segmentation task.
>
> Moreover, we additionally report the running time with NVIDIA TITAN X GPU on the classification task. For each sample, GET takes 0.020s time, while PointNet++, MeshCNN, GEM-CNN, HSN cost 0.037s, 0.083s, 0.190s, and 0.073s, respectively. So GET is more computationally efficient.
>
> ## Q6: Adding citations into Table 1 and 2 would be more readable.
>
> A6: We add the citations of all models in Table 1 and 2 here. And we will present this modification in the final revision.
>
> **Table 1**
>
> |     Model      | Rotation Aug. | Acc. (%) |
> | :------------: | :-----------: | :------: |
> |  MDGCNN [28]   |       √       |   82.2   |
> |    GI [36]     |       √       |   88.6   |
> |   GWCNN [16]   |       √       |   90.3   |
> |  MeshCNN [19]  |       ×       |   91.0   |
> |    HSN [46]    |       √       |   96.1   |
> | **GET (Ours)** |       ×       | **99.2** |
>
> **Table 2**
>
> |      Model      | Rotation Aug. | Acc. (%) |
> | :-------------: | :-----------: | :------: |
> |   MDGCNN [28]   |       √       |   89.5   |
> | PointNet++ [31] |       √       |   90.8   |
> |    HSN [46]     |       √       |   91.1   |
> |   PFCNN [49]    |       ×       |   91.5   |
> |  MeshCNN [19]   |       ×       |   92.3   |
> | **GET (Ours)**  |       ×       | **92.6** |
>
> ## Q7: The gauge equivariance is not strict (up to error bound). Is the title will be misleading?
>
> A7: No. (1)Many other gauge equivariant models [12, 28] are also only equivariant approximately. (2) We have proved in thm2,  as we increase N to large enough, the gauge equivariance error can be negligible.

---

> > ### Comment · Reviewer_3RJ5 · 2021-08-24
> > **Reply to the feedback**
> >
> > Thanks a lot for the further clarification and additional experiments. I think they addressed my concerns.

---

> > > ### Author Response · Authors · 2021-08-25
> > > **Thanks and kind reminder**
> > >
> > > Thanks for the valuable comments. We have noticed that you appreciate the addressing points in our feedback, but do not change the overall rating. This is just a kind reminder as you might be too busy and forget to update it. Thank you again for appreciating our work.

---

> > > > ### Comment · Reviewer_3RJ5 · 2021-08-31
> > > > **Re: authors**
> > > >
> > > > Hi,
> > > >
> > > > Thanks for the reminder!
> > > >
> > > > I will make the final decision before the discussion period ends. Please stay tuned. Because there are possibly internal discussions.
> > > >
> > > > All the best,
> > > >
> > > > 3RJ5

---

> > > ### Author Response · Authors · 2021-08-31
> > > **Is it possible that you update your score**
> > >
> > > Dear Reviewer 3RJ5,
> > > Many thanks for your extra time on reading our rebuttal! Now that you thought that our rebuttal had addressed your concerns, is it possible that you update your score? We understand that you may be very busy recently. Many thanks!

---

> > ### Author Response · Authors · 2021-09-02
> > **Thanks for supporting**
> >
> > Thanks for your support of our work and the improvement of the score.

---

### Official Review · Reviewer_5GYK · 2021-07-16

**Rating:** 8
**Confidence:** 4

**Summary:**

The authors address an important gap in existing literature at the intersection of self-attention networks (transformers) and the equivariant neural networks -- how to apply self-attention to manifolds or meshes (discretized versions of manifolds). This has immediate applications in the area of 3D shapes which are 2D manifolds embedded in 3D. The key challenge is that of gauge-equivariance as there is no canonical coordinate system and we would like the self-attention blocks to be equivariant to the choice of the gauge. To this end, the authors build a provably (approximate) gauge-invariant score function and a gauge-equivariant value function. The experiments show the importance of the proposed methods.

**Limitations And Societal Impact:**

The authors have not explicitly addressed the limitations of the work and they say that it's because of space limitations.

**Main Review:**

There are many strengths in this paper.
- The authors address an important and challenging problem and effectively advance research in applying self-attention networks on manifolds. The novelty in terms of ideas as applied to this problem appears to be high.
- They provide an intuitive generalization of the multi-head self-attention (MHSA) for feature-fields on manifolds.
- MHSA involves computing SA for multiple heads and then linearly combining them. However, the numerical values of the features depend on the gauge which in turn depends on the point on the manifold. Thus in order to compute pairwise self-attention scores at a point p, we need to first transform the numerical values features at a point q to the same gauge as at p and then use parallel transport to move them to p. The various weight matrices themselves need to satisfy constraints such that the score function is gauge-invariant and the value function is gauge-equivariant. This is challenging and the authors do a good job explaining and designing these modules.
- Theoretical guarantees ensure the desired gauge-equivariance of MHSA.
- The experiments clearly show the usefulness of the proposed approach. The number of trainable parameters to achieve the same performance as methods like MeshCNNs for shape classification and segmentation is reduced by an order of magnitude.

Points of clarification:
- I don't fully follow some of the notation and ideas in Section 4.1:
   - Why is the subscript $u$ needed in $q_u$?
   - And shouldn't the value function and the corresponding weight matrix be $V_q$ and $W_q$ instead of $V_u$ and $W_u$? This is because $u$ refers to a point using the gauge at point $p$, but the the weight matrix for the value function acts on the parallel transported point $q_u$.

The authors should discuss possible limitations of this work.

Table 4 suggests that the attention mechanism itself is not crucial for good performance. Can the authors comment on this?
Also, what is this baseline exactly which uses equivariance and not attention? The authors say that this is a convolution based model created by removing the attention scores. How does removing the attention scores make it convolutional. And is the resulting model similar to gauge-equivariant mesh CNNs?

**Time Spent Reviewing:**

6 hours

---

> ### Author Response · Authors · 2021-08-10
> **Initial Response**
>
> We sincerely thank Reviewer 5GYK for appreciating the novelty of this paper and enjoying the model design.
>
> Now we will answer to all of your questions, suggestions, comments and concerns:
>
>
> ## Q1: Why is the subscript $u$ needed in $q_u$?
>
> A1: $u=(u_1,u_2)^T\in \mathbb{R}^2$, $q_u=\exp_p w_p(u)$ is used in our notations. With the center point $p$, $u$ is the coordinate value under the gauge of point $p$ (*i.e.*, $w_p$), and can be comprehended as a point on the tangent plane $T_pM$. The Riemannian Exponential Map parameterizes local neighborhoods on manifolds, taking a relative coordinate $u$ as input and outputs a point on the manifold, say $q_u$. As the coordinate $u$ varies on the tangent plane $T_pM$, the output $q_u$ varies on the local neighborhood of the manifold $M$. For a specific $u$, $q_u$ indicates that this point on the manifold is obtained (parameterized) by this coordinate $u$.
>
> ## Q2: Shouldn't the value function and the corresponding weight matrix be $V_q$ and $W_q$ instead of $V_u$ and $W_u$?
>
> A2: As is discussed in Q1, $u$ embeds the local relative position information. Once the center point and the gauge is specified, each point in the neighborhood can be uniquely determined (parameterized) by the coordinate vector $u$. The value encoder matrix $W_V(u)$ is proposed to embed the relative position information regardless of the central point and the gauge, meaning that parameter sharing across points. $V_u$ is of the same logic, with the subscript indicating the local position information $u$.
>
> To clarify, $u$ is the coordinate vector and can be seen as a point on $T_pM$, not on the manifold $M$; The value encoder matrix $W_V(u)$ acts on $u$; And the value function $V_u$ is the matrix-vector product with the matrix $W_V(u)$ and the parallel transported feature vector ${f}^\prime_w(q_u)$.
>
> Hope the above clarification can help you figure out.
>
> ## Q3: Attention mechanism itself is not crucial for good performance. Can the authors comment on this?
>
> A3: Our GET achieves the state-of-the-art performance on the shape segmentation task (92.6%), 0.3% better than the non-attentive baseline. Now we find that it is not always the case, when applying GET model and the non-attentive baseline to the shape classification task, they give the accuracies of 99.2% and 96.7%, respectively. This yields a significant performance gain (2.5%), showing the great effect of self-attention.
>
> As for the shape segmentation task, the reason why the performance improvement is not fully realized here is perhaps relevant to the labels of the neighboring points. Since each point in the local neighborhood is labeled, and in some cases, the labels are of several types, the attention machanism may have no obvious advantage comparing with convolution then.
>
> We make the first step towards applying principled Transformer on the manifold, and we believe that the research progress with respect to Transformers on Vision task would help facilitate extending and improving our model.
>
> ## Q4: What is this baseline exactly which uses equivariance and not attention?
>
> A4: The model is given by setting all the attention scores of GET to 1, which is clearly equivariant but not attentive, shown as the baseline 2 in Table 4.
>
> ## Q5: Are the resulting model similar to gauge equivariant mesh CNNs?
>
> A5: Yes, the resulting one is similar to gauge equivariant mesh CNNs for they are both linear gauge equivariant models. But they differ in the value encoder matrix, the way to parallel transport, the global rotation invariance property, and the remained multi-head architecture.
>
> ## Q6: Limitations
>
> A6: Due to space limitations, we do not discuss limitations in the main paper. Here we address two possible limitations.
>
> **Partial Application.** GET can only be applied to orientable surfaces, as we have presumed the orientation of parameterization for each point on the surface, so we only consider the SO(2) transformations among different gauges, while, for unorientable surfaces, such as the Möbius strip, it is impossible to presume such an orientation.
>
> **Extra Hyperparameters.** GET introduces several additional hyperparameters, such as the order of the group $C_N$ and the order of Taylor Expansion, calling for extra time and energy in hyperparameter tuning when in implementation.

---

> > ### Comment · Reviewer_5GYK · 2021-08-26
> > **Thank you for the response**
> >
> > I thank the authors for answering my questions.
> >
> > I have read the other reviews and the authors' comments to those reviews. I largely agree with the reviews with positive scores. I believe the issues brought up by Reviewer 3RJ5 have been addressed well. I am not too bothered that the paper does not show exact gauge equivariance in practice.
> >
> > I am keeping my score as is and vote for acceptance.

---

### Official Review · Reviewer_apqm · 2021-07-20

**Rating:** 7
**Confidence:** 3

**Summary:**

This paper proposes a gauge-equivariant transformer (GET) for manifolds. The challenging issue for self-attention on a manifold is the lack of canonical coordinate systems on a surface, which results in rotation ambiguity. To address it, this work introduces the regular field of cyclic groups as feature fields, proposes a novel parallel transport approach that puts the feature vectors in these fields, and develops the gauge-equivariant self-attention by adopting Taylor expansion in solving equivariant constraints. The proposed method is validated on the tasks of 3D shape classification and segmentation, achieving SOTA on SHREC dataset and the Human Body Segmentation dataset.

**Ethical Concerns:**

I don't find any concerns.

**Main Review:**

Strength
+ This work is the first for gauge-equivariant self-attention (transformer).
+ The paper is well written with a solid background and the related properties being proven.
+ The technical contribution is high; a new parallel transport method by extending regular representation,  gauge equivariant value function by Taylor approximation.
+ The proposed method achieves SOTA on two popular benchmarks.

Weakness
- Unclear parts. The design of the gauge-invariant attention (Eq.19) needs to be clarified. Why not use the conventional product-based score?
- Lack of comparison to group-equivariant transformers.
While group-equivariant transformers [17, 21, 34] are mentioned in the related work, any of them are not compared in the experiments. The benefit of gauge equivariance would be better shown in comparison to the work of group equivariance. I think at least the work of [17] can be compared since they are developed for 3D data.
- Lack of in-depth comparison with gauge-equivariant CNNs.
Table 4 shows that the convolution-based gauge-equivariant network performs only slightly worse than GET. Does it imply the marginal gain of the proposed self-attention layer compared to gauge-equivariant convolution? In this sense, a more in-depth comparison with gauge-equivariant CNNs [10, 46, 12] needs to be done to show the advantage of GET.



**Time Spent Reviewing:**

7

---

> ### Author Response · Authors · 2021-08-10
> **Initial Response**
>
> We sincerely thank Reviewer apqm for appreciating the contributions and novelty of this paper, as well as the writing structure and experiment results.
>
> Now we will answer to all of your questions, suggestions, comments and concerns:
>
>
> ## Q1: Why not use the conventional product-based score?
>
> A1: We have experimented with the product-based score function in our experiments which exhibit an inferior performance than the gauge equivariant CNN based method and we conjecture that this is attributed to the optimization difficulty of the structure in our setting. So we propose Eq.19, following the design in the "Graph Attention Networks" [39] which results in an improved performance than gauge equivariant CNN.
>
> ## Q2: Comparison to group-equivariant Transformers
>
> A2: Group equivariance focuses on the equivariant property on transformations of **global** coordinate systems with respect to certain groups, while gauge equivariance focus on transformations of **local** coordinate systems. Also, group equivariance models basically treat the data samples as point clouds (in 3D), and gauge equivariance models treat them as 2D manifolds. Given the prior knowledge that the data is of the mesh type, which is the discretization of 2D manifold, it is reasonable that gauge equivariant models outperform group equivariant ones.
>
> In addition, we apply the SE(3)-Transformer [17] on the Human Body Segmentation dataset. SE(3)-Transformer achieves 91.8% accuracy in the shape segmentation task, inferior to GET which reaches 92.6% accuracy.
>
> ## Q3: In-depth comparison with gauge-equivariant CNNs
>
> A3: Gauge Equivariant Convolutional Networks and the Icosahedral CNN [10] is the seminal work of gauge equivariant networks, implemented on regular grids. Gauge Equivariant Mesh CNN (GEM-CNN) [12] and HSN [46] are the successful extensions of [10] on general meshgrids. GET differs from [12] and [46] mainly in follows:
>
> (1) GET uses self-attention to aggregate local feature information, while [12] and [46] are based on convolution.
>
> (2) The input of GET is projected local position vector which is treated as vector field, enabling GET to be globally rotation invariant. [12] and [46] directly adopt the global position vector as input, treated as scalar field, thus lose the global rotation invariance property.
>
>
> (3) GET adopts regular field as the type of intermediate feature fields, while the other two adopt the irreducible representation feature type. Regular field often gives better performance, and the reasons are discussed in Reviewer 3RJ5 Q4.
>
> In addition, we provide the results of GEM-CNN and HSN in both two tasks. In the classification task, GEM-CNN gives 93.3% accuracy and HSN gives 96.1% accuracy; In the segmentation task, GEM-CNN gives 91.3% accuracy and HSN gives 91.1% accuracy. All of the results are inferior to ours, which are 99.2% on the classification task and 92.6% on the segmentation task, respectively.
>
> ## Q4: Does it imply the marginal gain of the proposed self-attention layer compared to gauge-equivariant convolution?
>
> A4: Our GET achieves the state-of-the-art performance on the shape segmentation task (92.6%), 0.3% better than the non-attentive baseline. Now we find that it is not always the case, when applying GET model and the non-attentive baseline to the shape classification task, they give the accuracies of 99.2% and 96.7%, respectively. This yields a significant performance gain (2.5%), showing the great effect of self-attention.
>
> As for the shape segmentation task, the reason why the performance improvement is not fully realized here is perhaps relevant to the labels of the neighboring points. Since each point in the local neighborhood is labeled, and in some cases, the labels are of several types, the attention machanism may have no obvious advantage comparing with convolution then.
>
> We make the first step towards applying principled Transformer on the manifold, and we believe that the research progress with respect to Transformers on Vision task would help facilitate extending and improving our model.

---

### Official Review · Reviewer_mwoh · 2021-07-21

**Rating:** 6
**Confidence:** 3

**Summary:**

This paper considers the problem of introducing attention to data lying on manifolds (without a global symmetry). Since there is no canonical coordinate system available in such cases to parameterize neighborhoods, the authors instead use the notion of gauge equivariance, proposing a so-called gauge equivariant transformer. Some additional innovations in the intermediate layers are introduced to improve expressivity, and equivariance error is characterized. Experiments are reported on SHREC and Human Body Segmentation tasks, showing good performance with significant parameter efficiency compared to prior art.

**Limitations And Societal Impact:**

Limitations are not discussed.

**Main Review:**

This paper is concerned with the introduction of attention for operating on data lying on manifolds. General manifolds lack a canonical coordinate system to parameterize local neighborhoods. Since attention-based methods rely on such information for positional encodings, the paper considers a model that is gauge-equivariant (and thus agnostic to the orientation of the local coordinate system), introducing what the authors dub as a "Gauge Equivariant Transformer."

The starting point is the influential work on gauge-equivariant CNNs, at the core of which is a convolution like operation that is equivariant to local gauge transformations, thus providing principled manifold CNNs. However, the use of convolution also implies that all points in a neighborhood are accorded equal importance. This paper deals with the extension of this machinery to attention mechanisms -- such that the aggregation only depends on the intrinsic geometry. Further, an additional requirement pursued is global rotational invariance (which might not always be the case in gauge-equivariant CNNs for general manifolds). This work can also be considered an addition to a growing body of work on group-equivariant transformer models that have been proposed recently.

Attention is restricted to 2-manifolds (embedded in 3D). The basic definitions and preliminaries are introduced in section 3.1, where quantities such as gauges, tangent vectors, feature fields are defined, followed by explication on gauge transformations. The condition for gauge equivariance is specified in section 3.2, while the exponential and log map are covered in 3.3. Since self-attention amounts to aggregation of neighborhood features, in a general manifold, feature vectors of different points reside in different spaces, and thus have to be parallel transported to the same feature space where they can be processed. This process is described in section 3.4. The usual self-attention formulation is described in 3.5.

Following the above background, a gauge equivariant self-attention layer is defined where the value function is required to be gauge-equivariant and the attention score gauge-invariant. The value function is taken to be the numerical outcome of the parallel transported feature vector times the value matrix, subject to the usual gauge equivariant condition. The resulting equation is then reduced to a system of linear equations using a Taylor expansion. Then, using a similar structure as the graph attention network, the key and query are combined to give a gauge-invariant attention score. Finally experimental results are shown on SHREC and Human Body Segmentation datasets (the manifolds are represented as triangular meshes), the results show that the proposed model gives superior performance while using a fraction of parameters (compared to the baseline models). Results are also shown for the choice of N (the order of the cyclic group, which is used for equivariance in the intermediate layers).

While I have tried to verify all the details in the paper, my chief concern is that the paper can sometimes be hard to parse (which is also a reason for my lower confidence score). So, even with quite some familiarity with the related literature, I don't feel fully confident about some aspects of the paper (such as solving for the gauge equivariant condition). Nonetheless, the overall formulation seems correct, and the appendix fills in a lot of the missing details. Despite these concerns, I think the paper presents good results on two benchmarks and makes a step towards more principled self-attention networks for data on manifolds.


Minor comments:

- The paper keeps using the phrase "orientation ambiguity problem", but doesn't quite explain what they mean by it, perhaps due to the presumption that since the paper is on gauge equivariance, it will be clear. However, ii would improve readability if this is clarified concretely in the context of attention right in the beginning.
- The paper has many grammatical errors and typos, which the authors might want to iron out. I am listing some such examples below.
- Line 34: "neglection to content-based information" --> Neglect of content-based information?
- Line 49: "which initiatively" is this a typo?
- Line 63: "emerging field concerning on adapting neural networks on various data types" --> "emerging field concerned with adapting neural networks to various data types"
- Line 64: "For researches" --> For research on modeling/for research works on modeling
- Line 160: "selectively concentrating on the most relevant parts" --> "selectively concentrate on the most relevant parts"
- Line 211: "feature vector multiply by the value encoding matrix" --> "feature vector multiplied by the value encoding matrix"
- Line 261: "the resulted one will be" --> result will be

**Time Spent Reviewing:**

3.5 hours

---

> ### Author Response · Authors · 2021-08-10
> **Initial Response**
>
> We sincerely thank Reviewer mwoh for providing a comprehensive and detailed review, and affirming the superiority of the experiment results.
>
> Now we will answer to all of your questions, suggestions, comments and concerns:
>
> ## Q1: The orientation ambiguity problem
>
> A1: Unlike regular data, such as images, where each neighbor owns a clearly quantified relative position to its center in a canonical coordinate system, irregular data do not have a uniquely defined local coordinate system for the neighbors, resulting in the problem of orientation ambiguity, which directly obstructs the Transformer to numerically intake the relative position information.
>
> This is explained at the second paragraph in Section 1.
>
> ## Q2: Grammatical errors and typos
>
> A2: Line 34: The correct expression should be "Neglect of content-based information".
>
> Line 49: This is a typo. The correct expression should be "which".
>
> Line 63, 64, 160, 211, 261: they are some grammatical errors.
>
> Thanks for pointing out the problems. We will resolve them in the final revision.
>
> ## Q3: Limitations
>
> A3: Due to space limitations, we do not discuss limitations in the main paper. Here we address two possible limitations.
>
> **Partial Application.** GET can only be applied to orientable surfaces, as we have presumed the orientation of parameterization for each point on the surface, so we only consider the SO(2) transformations among different gauges, while, for unorientable surfaces, such as the Möbius strip, it is impossible to presume such an orientation.
>
> **Extra Hyperparameters.** GET introduces several additional hyperparameters, such as the order of the group $C_N$ and the order of Taylor Expansion, calling for extra time and energy in hyperparameter tuning when in implementation.

---

### Official Review · Reviewer_UDQP · 2021-07-22

**Rating:** 8
**Confidence:** 4

**Summary:**

This paper addresses the main challenge in attention to manifolds, namely, that there are no coordinates to use as input in positional encoding. The network uses regular fields of cyclic groups and a parallel transport method for transporting features to these fields. Invariance to global rotations is achieved by taking only local position vectors. Experiments are performed on SHREC and Human Segmentation using a fraction of the usual transformer capacity.




**Limitations And Societal Impact:**

See above.

**Main Review:**



This is the first gauge equivariant self-attention on manifolds.
Several contributions of the paper are novel: the parallel transport, the discretization of local rotations and their interpolation, and the approach for solving equivariant constraints.

The key in the authors' contribution are the cyclic fields obtained by the discretization of the local mesh angles.

Figure 3 is very illuminating in the parallel transport of feature vectors.

The computation of the value function follows the icosahedral CNN paper by Cohen.

Theorem 2 provides a bound with respect to any angle (not just the discretized ones).

The appendix is as useful to read as the paper and the derivations are a delight to follow.

The main puzzle for the reader is in the ablation study. Against the trend, it seems that it is mainly the gauge equivariance in human segmentation that matters rather than the self-attention (performance difference 0.3\%). The explanation though
"Without attention, the convolution-based baseline performs as well as the second best baseline (MeshCNN)" does not reflect the table. Please elaborate.

In a similar tone, please explain the challenges of gauge equivariance in self-attention rather than just gauge equivariance in meshes. After following the math and the lack of any difference in performance I am not sure whether the self-attention really matters and whether it is worth writing a paper about it.

POST-REBUTTAL: My main question was answered here and in the rest of the rebuttal. Stay with initial mark.

**Time Spent Reviewing:**

5

---

> ### Author Response · Authors · 2021-08-10
> **Initial Response**
>
> We sincerely thank Reviewer UDQP for appreciating the contributions and novelty of this paper.
>
> Now we will answer to all of your questions, suggestions, comments and concerns:
>
> ## Q1: 0.3% performance gain in the Human Body Segmentation dataset
>
> A1: Our GET achieves the state-of-the-art performance on the shape segmentation task (92.6%), 0.3% better than the non-attentive baseline. Now we find that it is not always the case, when applying GET model and the non-attentive baseline to the shape classification task, they give the accuracies of 99.2% and 96.7%, respectively. This yields a significant performance gain (2.5%), showing the great effect of self-attention.
>
> As for the shape segmentation task, the reason why the performance improvement is not fully realized here is perhaps relevant to the labels of the neighboring points. Since each point in the local neighborhood is labeled, and in some cases, the labels are of several types, the attention machanism may have no obvious advantage comparing with convolution then.
>
> We make the first step towards applying principled Transformer on the manifold, and we believe that the research progress with respect to Transformers on Vision task would help facilitate extending and improving our model.
>
> ## Q2: The second best baseline MeshCNN does not reflect the table
>
> A2: The convolution-based baseline is the Baseline 2 in Table 4, and the second best baseline is the MeshCNN model in Table 2. Both yield 92.3% accuracy, so they perform equally well.
>
> ## Q3: The challenges of gauge equivariance in self-attention
>
> A3: It is not trivial to develop gauge equivariance in attention, although we already have various gauge equivariant convolutional networks.
> An attention score comprises key, query and score function of the feature vectors, and the numerical value of the feature vectors vary among the choices of gauges.
> Simply adding an attention score to CNNs will incur the loss of equivariance property, largely jeopardizing the model performance. So the challenge lies in developing an equivariant key function, an equivariant value function and an invariant score function that make the attention score gauge invariant, and a value function which is gauge equivariant.

---

### Decision · Program_Chairs · 2021-09-27

**Decision:**

Accept (Poster)

**Comment:**

This paper develops the tools necessary to construct Gauge-equivariant transformers -- and to apply self-attention to manifolds or meshes with a focus on 2D smooth and orientable manifolds.
The presented methods can be applied to a wide variety of problems.


This paper is clearly written and technically sound. The experiments clearly show the strength of the proposed method (e.g. in comparison to MeshCNNs for shape classification).

Experimental results (Tables 1, 2) also support the idea that directly incorporating equivariances and invariances into the model's architecture is more efficient than using data-augmentation.